# Spiking activity in the visual thalamus is coupled to pupil dynamics across temporal scales

**Davide Crombie**[1,2], **Martin A. Spacek**[1], **Christian Leibold**[1,3‡]*, **Laura Busse**[1,4‡]*

1 Division of Neuroscience, Faculty of Biology, LMU Munich, Munich, Germany, 2 Graduate School of Systemic Neurosciences, LMU Munich, Munich, Germany, 3 Fakultät für Biologie & Bernstein Center Freiburg, Albert-Ludwigs-Universität Freiburg, Freiburg im Breisgau, Germany, 4 Bernstein Center for Computational Neuroscience, Munich, Germany

‡ These authors share senior authorship to this work.
* christian.leibold@biologie.uni-freiburg.de (CL); busse@bio.lmu.de (LB)

**Data Availability Statement:** The data and code required to generate the results and figures presented in this manuscript are available at https://doi.org/10.12751/g-node.ls6w8h.

## Abstract

The processing of sensory information, even at early stages, is influenced by the internal state of the animal. Internal states, such as arousal, are often characterized by relating neural activity to a single "level" of arousal, defined by a behavioral indicator such as pupil size. In this study, we expand the understanding of arousal-related modulations in sensory systems by uncovering multiple timescales of pupil dynamics and their relationship to neural activity. Specifically, we observed a robust coupling between spiking activity in the mouse dorsolateral geniculate nucleus (dLGN) of the thalamus and pupil dynamics across timescales spanning a few seconds to several minutes. Throughout all these timescales, 2 distinct spiking modes—individual tonic spikes and tightly clustered bursts of spikes— preferred opposite phases of pupil dynamics. This multi-scale coupling reveals modulations distinct from those captured by pupil size per se, locomotion, and eye movements. Furthermore, coupling persisted even during viewing of a naturalistic movie, where it contributed to differences in the encoding of visual information. We conclude that dLGN spiking activity is under the simultaneous influence of multiple arousal-related processes associated with pupil dynamics occurring over a broad range of timescales.

## Introduction

Information processing, even at the earliest sensory stages, can be modulated by several influences. One prominent influence is that of arousal-related behavioral states, which have been shown to change neural activity throughout the brain [1–4]. One classic indicator for arousal is pupil size, a metric that is relatively simple to measure and analyze and that has provided fundamental insights into neuromodulatory and cognitive influences on brain activity [2–8]. However, a multitude of factors converge to affect the pupil size signal, and how they are combined into this single indicator of arousal is not known. Likewise, how the various influences on the pupil signal relate to spontaneous and stimulus driven neural activity is not well understood.

**Funding:** This research was supported by the German Research Foundation (DFG) RTG2175 "Perception in context and its neural basis" (CL); DFG SFB 870 "Assembly and function of neural circuits", TP 19 (118803580) (LB); DFG SFB 1233 "Robust Vision: Inference Principles and Neural Mechanisms", TP 13 (276693517) (LB); ONE Munich Strategy Forum grant (LMU Munich, TU Munich and the Bavarian Ministry for Science and Art) (LB) and by the SmartStart program of the Bernstein Network for Computational Neuroscience funded by the Volkswagen Foundation (DC). None of the sponsors or funders had any role in the study design, data collection and analysis, decision to publish, or preparation of the manuscript.

**Competing interests:** The authors have declared that no competing interests exist.

**Abbreviations:** BF, basal forebrain; CPD, components of pupil dynamic; dLGN, dorsal lateral geniculate nucleus; EMD, empirical mode decomposition; LC, locus coeruleus; LCD, liquid crystal display; SVC, support-vector classifier; RVR, response variability ratio.

An ideal system to investigate arousal-related modulations of neural activity and sensory processing is the visual thalamus. The thalamic dorsal lateral geniculate nucleus (dLGN) is the primary relay of visual signals from the retina to the visual cortex, and dLGN neurons, like those in other thalamic nuclei, have long been known to display prominent patterns of activity associated with arousal [9–12]. In particular, 2 state-related firing modes have been described: burst firing, which is more prevalent during low-arousal states [9] and behavioral inactivity [12,13], and tonic firing, which is observed during alertness. Burst firing in the thalamus is characterized by a high frequency discharge of action potentials after sustained hyperpolarization, and it relies on the activation of low-threshold, transient (T-type) calcium channels (reviewed in [14]). In contrast, tonic firing occurs when the membrane potential is relatively depolarized, and T-type calcium channels are inactivated [14]. Since thalamorecipient circuits in primary sensory cortices are highly sensitive to the temporal coordination of inputs [15–17], the presence of bursts and tonic firing can have different effects on postsynaptic cortical neurons [18]. This has led to the hypothesis that thalamic nuclei use burst and tonic firing modes to gate or alter the flow of information to and between cortical areas according to the arousal state of the animal [14].

While previous studies have often relied on a single variable to define the state of the animal, several lines of evidence suggest that arousal-related modulations of neuronal activity cannot be adequately characterized by assigning mutually exclusive states to experimental epochs. For example, studies simultaneously measuring locomotion and pupil-linked arousal have revealed distinct effects of each in the visual cortex [6,8]. More generally, a substantial fraction of shared variability in visual cortex activity can be explained by high-dimensional sets of behaviors [19], suggesting that the state of sensory systems at any given moment results from a combination of multiple processes [3]. One potential factor that may distinguish these processes is the timescale over which they extend. For example, single arousal-related neuromodulators can have an impact across several timescales [20–23], and multiple neuromodulatory systems can influence neural activity over broadly different timescales [24].

To move beyond relating neural modulations to mutually exclusive states of arousal, we characterized modulations of spiking activity in the dLGN with respect to pupil size dynamics, taking into account that arousal-related processes occur across a wide range of temporal scales. We discovered that both tonic and burst spiking in the dLGN were coupled to fluctuations in pupil size over timescales ranging from seconds to minutes. Across these timescales, tonic spikes preferred opposite phases of the pupil signal compared to bursts. These multi-scale pupil dynamics captured modulations of dLGN activity beyond those explained by the pupil size per se and could occur in the absence of changes in locomotion state or saccadic eye movements. Furthermore, these modulations were also prevalent during presentation of naturalistic movies, despite the presence of rich stimulus-driven neural activity. Finally, we found that opposing phases of pupil dynamics across all timescales were associated with differences in the encoding of naturalistic movies by the dLGN, indicating that pupil-linked neural activity modulations across various timescales contribute to state-dependent differences in the flow of sensory information to the cortex. Our findings support the notion that arousal-related modulation, rather than being a singular process, likely involves an interplay of changes occurring over diverse timescales.

## Results

To assess how dLGN spiking activity is influenced by internal state, we paired extracellular silicon probe recordings with video-based analysis of pupil size. During these recordings, mice were head-fixed, but free to run on an air-cushioned styrofoam ball while viewing a static gray

screen or a sparse noise stimulus (Fig 1A). We measured pupil size and locomotion speed as indicators of the internal and behavioral states of the animal. Throughout these recordings under largely isoluminant conditions, pupil size, a marker for internal states such as arousal, was in continuous fluctuation (see example in Fig 1B, black trace and $S1A_1$ Fig, gray and pink; $N = 15$ recording sessions in 10 mice). Similarly, active and quiescent behavioral states, as measured by locomotion (Fig 1B, green trace), were observed in all experiments. The proportion of time spent in locomotion varied across recordings, ranging from 0.11 to 0.47 (median = 0.27; $S1B_1$ Fig). Consistent with previous studies [8,12,25], bouts of locomotion were often accompanied by increases in pupil size ($S1B_4$ Fig). However, we also noticed that the fluctuations in pupil size were generally similar between active and quiescent behavioral states ($S1B_5$ Fig). In accordance with previous results [6,7], this suggests that diverse internal states may coexist within a behavioral state.

We observed that fluctuations in pupil size during both locomotion and quiescence were accompanied by changes in the spiking activity of dLGN neurons. For example, the neuron shown in Fig 1B generally increased its firing rate when the pupil was large. To characterize the relationship between pupil size and firing rates, we binned the spiking activity of individual dLGN neurons in 250 ms windows and examined mean spike counts across pupil sizes. We observed that the relationship between pupil size and dLGN firing rate varied between neurons, and was often non-monotonic ($Fig 1C_1$), reminiscent of previous observations in cortex [7]. Indeed, among the 89.7% of recorded dLGN neurons showing significant modulation across pupil sizes (140/156 neurons; one-way ANOVA, $p \leq 0.05$), the majority (78/140) had their peak firing rate outside the top decile of pupil size. Indeed, peak firing rates were observed across the entire range of pupil sizes ($Fig 1C_1$). Similarly, bursts of spikes (see Fig 2A) were also linked to pupil size in most dLGN neurons ($Fig 1C_2$; 93/145), but, unlike overall firing rates, burst rates tended to be highest at the smallest pupil sizes (72/93; only 21/93 had non-monotonic modulation profiles). To assess the degree to which the relationship between pupil size and spiking activity originated from changes in retinal illumination, we repeated these analyses for experiments conducted in darkness. We found that even in darkness, firing rates and bursting were modulated across pupil sizes in the majority of dLGN neurons (firing rate: 89/94 neurons; burst rate: 44/83 neurons), with many neurons showing high firing rates when the pupil was large ($S1C_1$ Fig) and bursting when the pupil was small ($S1C_2$ Fig), suggesting that under these stimulus conditions the overall relationship between pupil size and spiking activity originates from nonvisual factors. We therefore conclude that firing rates and bursting depend on pupil size in the majority of dLGN neurons.

Returning to Fig 1B, beyond the relationship to pupil size per se, we also found instances where firing rate increases were coupled to dilating phases of the pupil dynamics (Fig 1B, marked by 1). These dilation-related firing rate changes could occur while the pupil was relatively constricted (Fig 1B, marked by 2) or dilated (Fig 1B, marked by 3). Indeed, even in darkness, for a given pupil size the firing rate variability was often larger than the mean (median Fano factor = 1.4; $W = 4.4 \times 10^3$, $p = 6.4 \times 10^{-16}$, $N = 94$ neurons; $S1C_3$ Fig), suggesting the presence of additional modulatory processes not captured by pupil size. We thus sought to develop a framework that could capture this breadth of modulations in dLGN spiking activity by focusing on the multi-scale dynamics of pupil size.

To gain a better understanding of these modulations of dLGN firing rates, we explored the multi-scale dynamics of the pupil signal, aiming to extract information about changes in internal state beyond the size of the pupil per se. Indeed, previous studies dating to the beginning of pupillometry have suggested that pupil dynamics can be a relevant indicator of arousal [26–28]. The relationship between pupil dynamics and internal states has also been explored in mouse visual cortex [6], where associations between neuromodulatory signaling and pupil

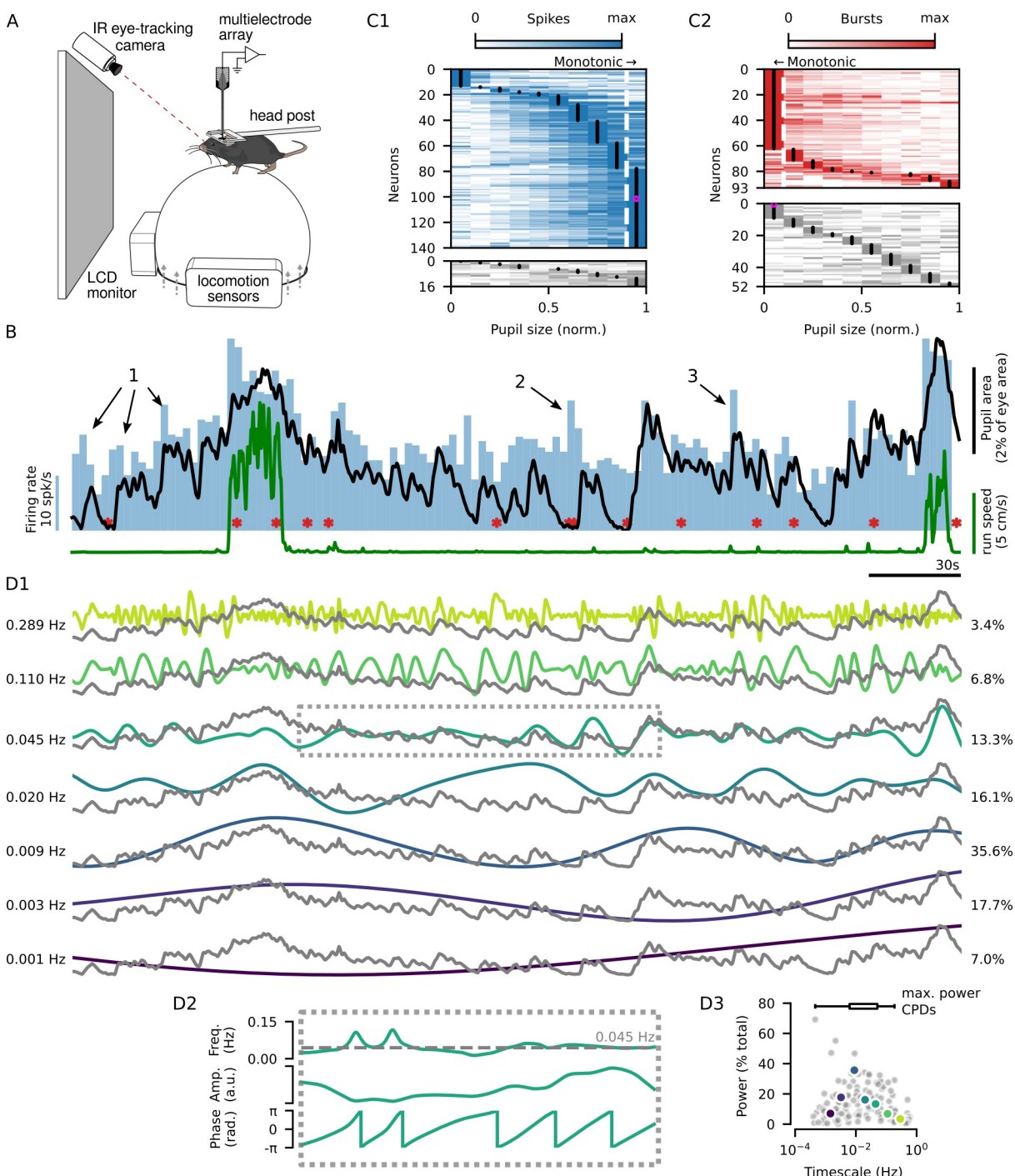

**Fig 1. Extracellular recordings from the dLGN reveal diverse arousal-related modulations of firing rate. (A)** Schematic of the experimental setup. **(B)** Pupil size (black) and locomotion speed (green) overlaid on the firing rate (blue, binned in 2.5 s windows) and spike bursts (red asterisks) from an example dLGN neuron. **(C1)** Spike counts (min-max normalized) across pupil sizes (min-max normalized) for 140/156 dLGN neurons with significant modulation (top, one-way ANOVA $p \leq 0.05$ across 10 pupil size bins) and 16/156 dLGN neurons without significant modulation (bottom). Neurons were sorted by the pupil size with the highest spike count (black dots). The example neuron from (B) is shown in magenta. The dashed white line indicates the 90th percentile of pupil size. **(C2)** Same as (C1) but for bursts of spikes. Neurons were sorted by the pupil size with the highest burst count. The dashed white line indicates the 10th percentile of pupil size. **(D)** The pupil size trace in (B) (gray traces) separated into CPDs (colored traces) occurring over different timescales. The components are described by their characteristic timescale (D1, left) and power (% total, D1, right), as illustrated for the trace indicated by the dashed box in (D2). **(D2)** Top: The characteristic timescale (gray dashed line) is computed as the amplitude-weighted mean of the component's instantaneous frequency. Middle: The component's power is defined as fraction of total power density, computed from the squared amplitudes of all CPDs derived from the

same pupil signal. Bottom: The phase of the CPD describes if the pupil is undergoing dilation ($-\pi$ to 0) or constriction (0 to $\pi$) at the timescale defined by the component. **(D3)** Gray dots: The characteristic timescale of each CPD plotted against its power for CPDs extracted from all recordings ($N$ = 14). Colored dots: Components from the example recording in (B, D). The inset box plot (top) shows the range and IQR of the components with the maximum power from each recording. CPD, components of pupil dynamics; dLGN, dorsal lateral geniculate nucleus.

dynamics were reported [24]. Building on these studies (see also [7]), we observed that pupil size dynamics could be described over a variety of temporal scales: from minutes-long changes to cycles of dilation and constriction lasting 10s of seconds, to quicker changes on the order of seconds. We therefore employed a data-driven approach, called empirical mode decomposition (EMD, see Materials and methods; [29]), to split the pupil signal without prior assumptions into components capturing its underlying dynamics (Fig 1D). Each component of pupil dynamics (henceforth, CPD) is described by its characteristic timescale (Fig 1D$_1$, left) and its relative power (Fig 1D$_1$, right). The characteristic timescale describes the average period of the pupil dilation-contraction cycles captured by the component (Fig 1D$_2$; see Materials and methods). The extracted CPDs spanned several orders of magnitude in their characteristic timescale, capturing dilation-contraction cycles lasting from several minutes ($10^{-3}$ = 0.001 Hz) to just a second ($10^0$ = 1 Hz). Components with high power were found across this entire range, underscoring the multi-component nature of pupil size dynamics (Fig 1D$_3$). Additionally, the broad distribution of the CPD timescale with the highest power from each recording (Fig 1D$_3$, top) illustrates the diversity of pupil dynamics across recording sessions. Importantly, the set of CPDs extracted from a single pupil recording progress through dilation-contraction cycles largely independently from each other (S1D Fig; see Materials and methods), indicating that CPDs capture distinct aspects of pupil size dynamics.

Having captured pupil dynamics at multiple timescales, we went on to characterize their relationship to arousal-related dLGN activity. We separated the spiking activity from each dLGN neuron into tonic spiking and bursting (Fig 2A; bursts were defined as $\geq$2 spikes with $\leq$4 ms ISI preceded by $\geq$100 ms without spikes [30]). Consistent with previous findings in awake animals, we found that bursts were relatively rare, accounting for only 3% of spikes on average (median burst ratio for the 93% of neurons with bursting; S2A$_2$ Fig). Despite their small contribution to the total spike count (S2A$_1$ Fig), bursts provide an extracellular marker for the membrane potential status of thalamic neurons (indicating prolonged hyperpolarization) and play a role in determining rhythmic cortical states ([13]; see also S2B and S2C Fig). After separating burst and tonic spikes, we performed a phase coupling analysis for each neuron and simultaneously recorded CPD, collecting the phase of the CPD at the time of tonic spikes or bursts—considering all spikes in the burst as one single event (Fig 2B and 2C). The preferred phase of each spike type indicates whether they mainly occur during pupil dilation (-$\pi$ to 0) or contraction (0 to $\pi$) as captured by the CPD. Meanwhile, the coupling strength indicates the degree to which spikes and bursts adhere to the preferred phase and was computed with a bias-corrected metric that allows comparison between neurons with different rates of tonic spiking and bursting (see Materials and methods; S2A Fig). The statistical significance of the coupling was assessed using a permutation test that accounts for short-timescale spiking patterns that typically inflate coupling strength metrics (see Materials and methods; S2B Fig).

We found that both tonic spiking and bursts in dLGN neurons were coupled to the extracted components of pupil dynamics (CPDs) across a wide range of timescales (Fig 2C). Significant coupling of to at least 1 component of pupil dynamics was measured in 98.1% of dLGN neurons (153/156) for tonic spiking and 87.3% (110/126) for bursting. Examining the phase coupling for all neurons and simultaneously recorded CPDs (Fig 2C), we found that this

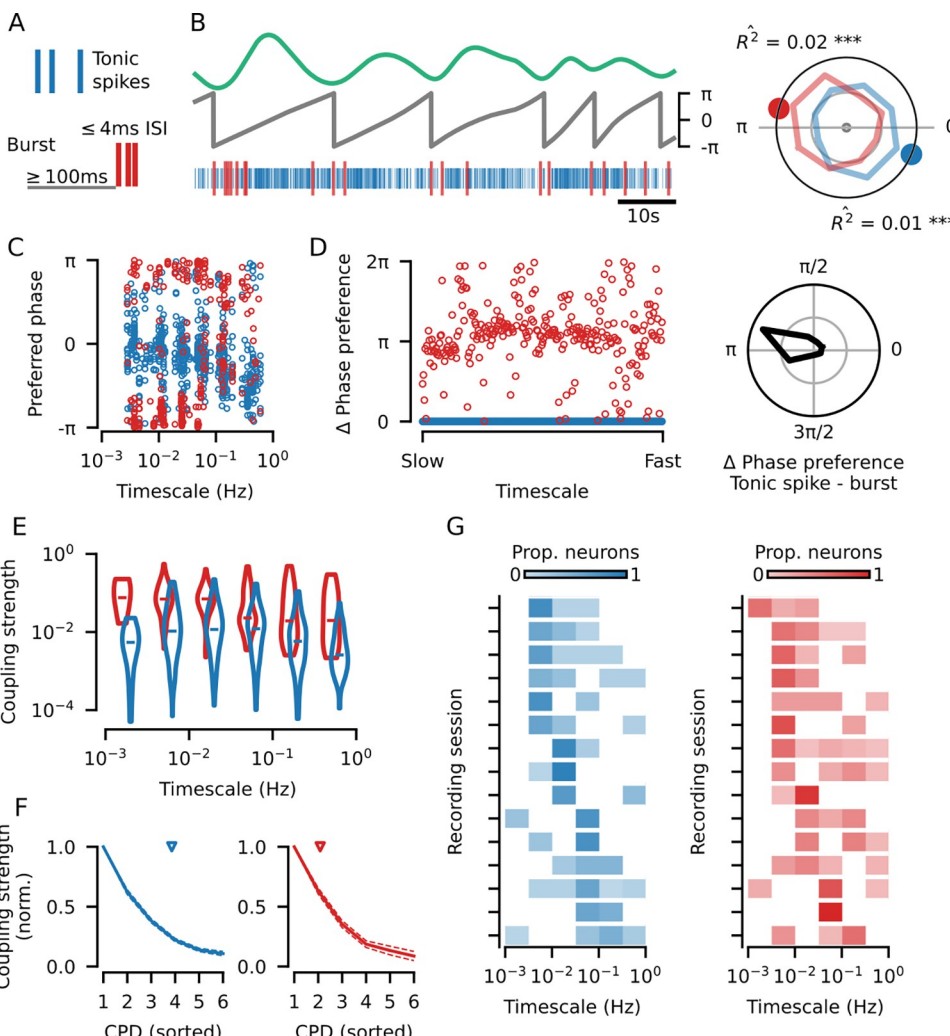

**Fig 2. Tonic spikes and bursts in the dLGN are coupled to pupil dynamics across multiple timescales. (A)** Bursts (red) were defined as $\geq 2$ spikes occurring after $> 100$ ms of silence and with an inter-spike interval less than 4 ms [30]; all remaining spikes were classified as tonic spikes (blue). For the phase coupling analysis below, all spikes in a burst were treated as a single burst event. **(B)** Top: example component of pupil dynamics (CPD). Middle: the corresponding phases of the example CPD. Bottom: simultaneously recorded tonic spiking (blue) and bursting (red) from an example neuron. Right: the phase distributions for bursts and tonic spikes from the example neuron-CPD pair ($\hat{R}^2$: coupling strength; colored dots: preferred phase). The statistical significance of coupling to a CPD was determined using a permutation test by shuffling 300 ms bins of spiking activity (see Materials and methods; asterisks: * for $p \leq 0.05$, ** for $p \leq 0.01$, *** for $p \leq 0.001$). **(C)** The preferred coupling phase of bursts and tonic spikes for all neuron-CPD pairs with significant coupling. Significant tonic spike coupling was observed in 98.1% of neurons (153/156) and burst coupling in 87.3% of neurons (110/126). **(D)** Left: Burst—tonic spike phase differences for each neuron-CPD pair in (C) with significant coupling for both types of spiking ($N = 284$ neuron-CPD pairs), sorted by CPD timescale. Right: distribution of burst—tonic spike phase differences (mean = 2.8, V-test for non-uniform distribution with a mean of $\pi$: V = 140.7, $p < 0.001$; grid lines indicate proportion of 0.25). **(E)** Coupling strength distributions for all significantly coupled neuron-CPD pairs, binned by CPD timescale (horizontal bars: median coupling strength; Kruskal–Wallis one-way ANOVA across timescale bins for tonic spike coupling strengths: H = 66.4, $p = 5.8 \times 10^{-13}$, $N = 681$ neuron-CPD pairs; burst coupling strengths: H = 32.1, $p = 5.8 \times 10^{-6}$, $N = 320$ neuron-CPD pairs; Wilcoxon rank-sum test for burst versus tonic spike coupling strengths: W = $3.3 \times 10^3$, $p = 2.0 \times 10^{-33}$, $N = 284$ neuron-CPD pairs). **(F)** Coupling strength decay across the multiple CPDs to which single neurons were coupled. Coupling strengths were measured after removal of periods of phase coupling between CPDs (S3B Fig; see Materials and methods) and normalized to the highest coupling strength for each unit (mean ± SEM; left: tonic spiking; right: bursting). Only neuron-CPD pairs with significant coupling after removal of CPD phase coupling were included (blue arrow: tonic spiking mean = 3.8 CPDs per neuron; red arrow: bursting mean = 2.1 CPDs per neuron). **(G)** Distribution of the preferred coupling timescale (CPD with the strongest coupling) of neurons recorded in each recording session for tonic spiking (left) and bursting (right). There is some variability in timescale preference between recording

sessions (one-way ANOVA for tonic spiking: F = 3.2, $p = 2.5 \times 10^{-4}$; bursting: F = 2.6, $p = 2.5 \times 10^{-3}$), sessions are sorted by the timescale to which most neurons had their strongest tuning. CPD, components of pupil dynamics; dLGN, dorsal lateral geniculate nucleus.

coupling was not limited to a single temporal scale, but rather occurred over time scales spanning seconds to several minutes (1 to 0.001 Hz). Across this wide range of time scales, tonic spikes, and bursts consistently preferred opposite phases of pupil dynamics (Fig 2D; mean tonic spike—burst preferred phase difference = 2.8; V-test for non-uniform distribution with a mean of $\pi$: V = 140.0, $p \leq 0.001$, N = 284 neuron-CPD pairs with significant coupling of both spike types). The coupling strengths for both tonic spiking and bursting differed across temporal scales (Fig 2E; Kruskal–Wallis one-way ANOVA for tonic spiking: H = 68.6, $p = 2.0 \times 10^{-13}$, N = 681 neuron-CPD pairs; bursting: H = 34.7, $p = 1.7 \times 10^{-6}$, N = 320 neuron-CPD pairs), although bursting consistently displayed stronger coupling than tonic spiking (Fig 2E; Wilcoxon rank-sum test: W = $3.4 \times 10^3$, $p = 7.1 \times 10^{-34}$, N = 284 neuron-CPD pairs), possibly because bursts are more exclusive to a certain membrane potential state of dLGN neurons. The same analyses performed on data collected in darkness also revealed phase coupling across temporal scales, with significant coupling observed in almost all neurons (S3A$_1$ Fig; tonic spiking: 96.8%, 91/94 neurons; bursting: 64.8%, 35/54 neurons). In darkness, the opposing phase preference between bursting and tonic spiking was largely preserved (S3A$_2$ Fig; mean tonic spike—burst preferred phase difference = 2.5; V = 25.8, $p = 5.1 \times 10^{-5}$, N = 88 neuron-CPD pairs with significant coupling of both spike types), and coupling was also consistently stronger for bursting than tonic spiking (S3A$_3$ Fig; Wilcoxon rank-sum test: W = 58.0, $p = 2.7 \times 10^{-15}$, N = 88 neuron-CPD pairs), together suggesting that coupling to CPDs was likely driven by changes in internal state rather than the changes in retinal illumination caused by pupil size fluctuation.

Next, we asked whether the coupling we observed across multiple temporal scales resulted from different neurons being modulated at different time scales, or if modulation at multiple temporal scales was present within the spiking of single neurons. Many neurons showed significant coupling to more than 1 CPD for both tonic spiking (mean = 4.4 CPDs with significant coupling per neuron) and bursting (mean = 2.5 CPDs per neuron). To ensure that potential phase relationships between the CPDs themselves (S1D Fig) did not underlie the observed coupling to multiple temporal scales, we repeated the phase coupling analysis after removing periods of time in which the components themselves are coupled (S3B Fig; see Materials and methods). Neurons retained their coupling to more than 1 temporal scale (tonic spiking mean = 3.8 CPDs per neuron; bursting mean = 2.1 CPDs per neuron), and coupling strengths across timescales remained unchanged for both tonic spiking (S3B$_3$ Fig; median = 0.0089, N = 563 neuron-CPD pairs versus original median = 0.0081, N = 682 neuron-CPD pairs; Mann–Whitney U test: U = $1.9 \times 10^5$, $p = 0.34$) and bursting (median = 0.0583, N = 250 neuron-CPD pairs versus original median = 0.0528, N = 320 neuron-CPD pairs; U = $3.9 \times 10^4$, $p = 0.56$). Among these neurons with multi-scale coupling, the gradual decay in coupling strengths from the strongest to the weakest indicated that modulations at the non-preferred timescales were not negligible (Fig 2F). Considering again the full dataset (Fig 2C–2E), the specific timescale to which a neuron was most strongly coupled was stable for individual neurons across subsamples of the data (S3D Fig) but varied between neurons such that strong coupling was observed across the entire range of timescales measured (Fig 2G). Notably, part of this variability could be attributed to the mouse and/or recording session (one-way ANOVA for tonic spiking timescale preferences across recordings: F = 3.2, $p = 2.5 \times 10^{-4}$; bursting: F = 2.6, $p = 2.5 \times 10^{-3}$), specifically, to the frequency of switches between locomotion and quiescence

in a given session (Pearson's R = 0.71, $p$ = 4.3x10$^{-3}$, $N$ = 14 recording sessions; S3C$_1$ Fig). However, in almost all recording sessions, we found the CPD associated with the strongest coupling (preferred timescale) of neurons from the same recording was distributed across more than 1 component (tonic spiking: 15/15 sessions; bursting: 14/15 sessions), indicating that the timescale of strongest modulation was not only influenced by factors common to the recording session, but also by neuron-specific factors. We thus conclude that the spiking activity of individual dLGN neurons can be coupled to pupil dynamics across multiple independent temporal scales, and that there is diversity in the timescale to which neurons are most strongly coupled.

Our phase coupling framework introduced above (Fig 2) not only captures modulations associated with aspects of pupil dynamics, but it is also capable of capturing modulations usually related to pupil size per se. For example, in Fig 1D, it is apparent that peaks and troughs of certain components coincide with large and small pupil sizes. To disentangle the influence of pupil size from CPD phase coupling, we used a subsampling approach to minimize differences in the distribution of pupil size between the phases of each CPD (Fig 3A; see Materials and methods). We then assessed phase coupling considering only the spikes that occurred in these subsampled periods, noting that, especially for slower CPDs, sometimes only a small proportion of the original data could be retained for analysis (median proportion retained (IQR) for timescales <0.1 Hz: 52% (39, 61); timescales >0.1 Hz: 82% (82, 88)). Despite this reduction in available data, we found that coupling to CPDs in most neurons was largely preserved for tonic spiking (significant coupling observed in 93.9% of neurons, 139/148; mean = 3.3 CPDs per neuron) and burst events, albeit in a smaller proportion (63.6% of neurons, 77/121; mean = 1.0 CPD per neuron). For neurons that retained phase coupling, the characteristic opposing phase preferences of bursts and tonic spikes within neurons was also preserved (Fig 3B; mean = −2.9; V-test for non-uniformity and a mean of $\pi$: V = 20.4, $p$ = 6.4 × 10$^{-4}$, $N$ = 80 neuron-CPD pairs). Overall, coupling strengths decreased for both tonic spikes (Fig 3C; overall median = 0.0081, $N$ = 681 neuron-CPD pairs versus size-matched median = 0.0046, $N$ = 489 neuron-CPD pairs; Mann–Whitney U test: U = 2.0 × 10$^5$, $p$ = 4.4 × 10$^{-7}$) and bursts (overall median = 0.0528, $N$ = 320 neuron-CPD pairs versus size-matched median = 0.0250, $N$ = 115; Mann–Whitney U test: U = 2.2 × 10$^4$, $p$ = 1.0 × 10$^{-3}$). This coupling strength decrease, however, was only present for a middle-range of temporal scales, suggesting that pupil size per se may not contribute to slow (timescale <0.03 Hz) or fast (>0.1 Hz) coupling in dLGN neurons. We then examined how the preferred timescale of individual neurons was affected after controlling for the influence of pupil size per se, finding that many neurons shifted the timescale to which they were most strongly coupled (Fig 3D, faded slices; tonic spikes: 55.9%; bursts: 30.2%). For bursting, such shifting of timescales was particularly prominent among neurons whose activity was monotonically related to pupil size (S3E Fig; Chi-squared test: $\chi^2$ = 9.5, $p$ = 8.7 × 10$^{-3}$; see also Fig 1C). Strikingly, however, we also found a sizeable proportion of neurons whose strongest coupling increased after controlling for pupil size (Fig 3D, highlighted slices; tonic spikes: 25.5%; bursts: 15.1%), suggesting that for these neurons effects related to pupil size per se were masking other firing rate modulations related to pupil dynamics. We thus conclude that the majority of dLGN neurons undergo multi-scale modulations of bursting and tonic spiking beyond those associated with pupil size per se.

We next investigated whether coupling between dLGN spiking and pupil size dynamics could be attributed to modulations driven by transitions in behavioral states. Previous studies have shown that changes in behavior, such as the transition from quiescence to locomotion, are accompanied by firing rate changes in dLGN neurons [12,31,32]. Consistent with these findings, we observed a decrease in tonic spiking in a 5 s window surrounding the offset of a locomotion bout (S4A$_1$ Fig; see also [8]), and a sharp increase in tonic spiking, preceded by a

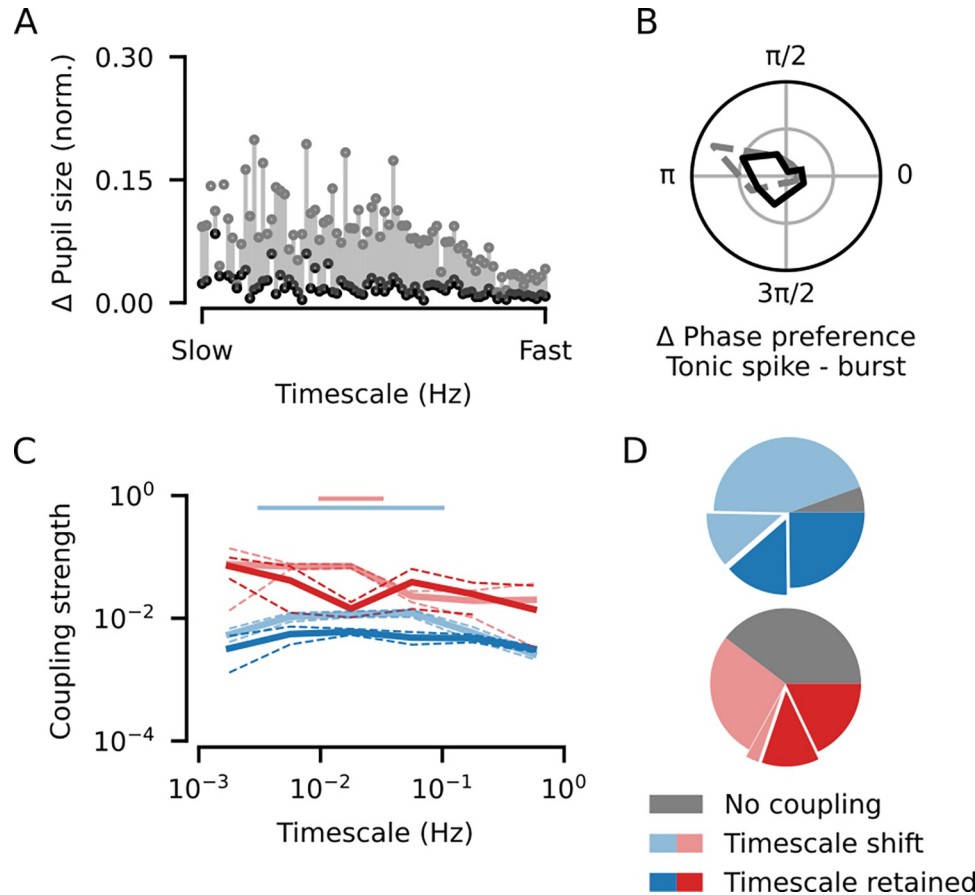

**Fig 3. Coupling of dLGN spiking to pupil dynamics persists after controlling for pupil size. (A)** Comparison of the difference in pupil size (min-max normalized) between the phase bins from each CPD sorted by CPD timescale before (gray) and after subsampling (black; see Materials and methods), illustrating the efficacy of the size matching procedure. **(B)** Distribution of burst—tonic spike phase differences measured after the size-matching procedure ($N$ = 80 neuron-CPD pairs; mean = −2.94, V-test for non-uniform distribution with a mean of $\pi$: V = 20.4, $p$ = 6.4 × 10$^{-4}$; gray: distribution from Fig 2D; grid lines indicate proportion of 0.25). **(C)** Coupling strengths (solid lines: median; dashed lines: bootstrapped SE of the median) measured before (faded lines; Fig 2E) and after (bold lines) the size-matching procedure for tonic spikes (blue; overall median = 0.0081, $N$ = 681 neuron-CPD pairs vs. size-matched median = 0.0046, $N$ = 489 neuron-CPD pairs; Mann–Whitney U test: U = 1.96 × 10$^5$, $p$ = 4.37 × 10$^{-7}$) and bursts (red; overall median = 0.0528, $N$ = 320 neuron-CPD pairs vs. size-matched median = 0.0250, $N$ = 115; Mann–Whitney U test: U = 2.22 × 10$^4$, $p$ = 1.04 × 10$^{-3}$). **(D)** Proportion of neurons with significant phase coupling from Fig 2 that kept the same preferred coupling timescale (bold slices; 38.6% for tonic spiking, 30.2% for bursting), shifted their preferred coupling timescale (faded slices; 55.9% for tonic spiking, 30.2% for bursting), or lost their coupling (gray slices; 5.5% for tonic spiking, 39.6% for bursting). The highlighted slices show the proportion of neurons that had an increase in coupling strength after controlling for pupil size (25.5% for tonic spiking, 15.1% for bursting). CPD, components of pupil dynamics; dLGN, dorsal lateral geniculate nucleus.

slight decrease in bursting, in a 4 s window surrounding the onset of a locomotion bout (S4A$_2$ Fig). Periods of locomotion were also associated with an increased pupil size (S1B$_4$ Fig). Unsurprisingly then, 55.3% of CPDs (52/94) had a small positive correlation with locomotion speed (S4B Fig; permutation test: $p \leq 0.05$, see Materials and methods). Given these findings, we asked if the CPD-linked spiking modulations we observed might be driven by these loco-motion-correlated components. As a first step, we compared phase coupling across components, and observed that significant coupling for tonic spikes was equally likely regardless of whether the component was correlated to locomotion or not (neuron-CPD pairs with signifi-cant coupling: correlated = 73.5% (305/415) versus uncorrelated = 75.9% (341/449); Chi-

squared test: $\chi^2 = 0.56$, $p = 0.45$). For bursting, locomotion-correlated components were even less likely to have significant coupling (correlated = 38.2% (128/335) versus uncorrelated = 50.3% (186/370); Chi-squared test: $\chi^2 = 9.9$, $p = 1.7 \times 10^{-3}$). Thus, while locomotion can drive changes in spiking activity, and locomotion speed is partially reflected in certain CPDs, these relationships do not appear to be the sole cause of coupling to pupil dynamics.

To explicitly remove the influence of behavioral state changes on the coupling between pupil dynamics and spiking activity, we computed the phase coupling taking into account only the spiking activity that occurred within a given behavioral state. We also excluded spiking activity in the transitional windows between behavioral states (S4A Fig; quiescence to locomotion: −2 to 2 s; locomotion to quiescence: −1 to 4 s). We first focused on periods of quiescence (Fig 4A$_1$), which constituted on average 72% of the recordings (median proportion of time outside of locomotion bouts; S1B$_1$ Fig). During these quiescent periods, where variance in locomotion speed was low (S1B$_3$ Fig), we found that the coupling between spiking activity and pupil dynamics was largely the same as when measured across the entire recording (Fig 4A$_2$; significant tonic spike coupling in 97.3% (143/147); burst coupling in 77.1% (84/109) of neurons). Neurons retained the anti-phase relationship between tonic spiking and bursting (Fig 4A$_3$; mean tonic spike—burst phase difference = 2.9, $N = 221$ neuron-CPD pairs; V-test for non-uniform distribution with a mean of $\pi$: V = 111.1, $p < 0.001$), with coupling to multiple timescales of pupil dynamics (tonic spiking mean = 4.2 CPDs per neuron; bursting mean = 2.3 CPDs per neuron), and similar coupling strengths across all timescales compared to the coupling measured across the whole recording for tonic spiking (Fig 4A$_4$; quiescence median = 0.0100, $N = 591$ neuron-CPD pairs versus overall median = 0.0081, $N = 681$ neuron-CPD pairs; Mann–Whitney U test: U = $1.9 \times 10^5$, $p = 2.6 \times 10^{-2}$) and bursting (quiescence median = 0.0564, $N = 256$ versus overall median = 0.0528, $N = 320$ neuron-CPD pairs; Mann–Whitney U test: U = $3.9 \times 10^4$, $p = 0.28$).

Having shown that phase coupling across timescales persisted when measured only during periods of quiescence, we next repeated the same analyses focusing on periods with locomotion bouts (Fig 4B$_1$) and found similar coupling characteristics as for the other states (Fig 4B$_2$; significant tonic spike coupling in 85.3% (122/143), burst coupling in 77.9% (60/77) of neurons). Tonic spikes and bursts showed preferences for opposing phases (Fig 4B$_3$; mean tonic spike—burst phase difference = 3.1, $N = 87$ neuron-CPD pairs; V-test for non-uniform distribution with a mean of $\pi$: V = 56.7, $p < 0.001$). There was a notable increase in coupling strengths for both tonic spikes (Fig 4B$_4$; locomotion median = 0.0254, $N = 359$ neuron-CPD pairs; Mann–Whitney U test: U = $7.2 \times 10^4$, $p = 2.6 \times 10^{-28}$) and bursts (locomotion median = 0.0927, $N = 133$ neuron-CPD pairs; Mann–Whitney U test: U = $1.3 \times 10^4$, $p = 6.2 \times 10^{-10}$). Given the higher locomotion speed variability during locomotion bouts (S1B$_3$ Fig), we hypothesized that particularly strong dLGN modulation in this state might be linked to these changes in overt behavior. We therefore identified CPDs that were correlated to locomotion speed only within bouts of locomotion (64.9% of CPDs, 61/94; S4C Fig). Consistent with this hypothesis, we found that the locomotion-correlated components were more likely to drive significant phase coupling for both tonic spikes (neuron-CPD pairs with significant coupling: correlated = 77.2% (477/618), uncorrelated = 68.7% (169/246); Chi-squared test: $\chi^2 = 6.3$, $p = 0.01$) and bursts (correlated = 49.6% (256/516), uncorrelated = 30.7% (58/189); Chi-squared test: $\chi^2 = 19.3$, $p = 1.1 \times 10^{-5}$). Tonic spike coupling strengths were also higher for locomotion-correlated components (correlated median = 0.0099, $N = 477$ versus uncorrelated median = 0.0054, $N = 169$; Mann–Whitney U test: U = $4.6 \times 10^4$, $p = 7.1 \times 10^{-3}$). Thus, while transitions between quiescence and locomotion bouts do not drive dLGN-pupil phase coupling, movement-related modulations may underlie a particularly strong coupling between dLGN spiking and pupil dynamics within periods of locomotion.

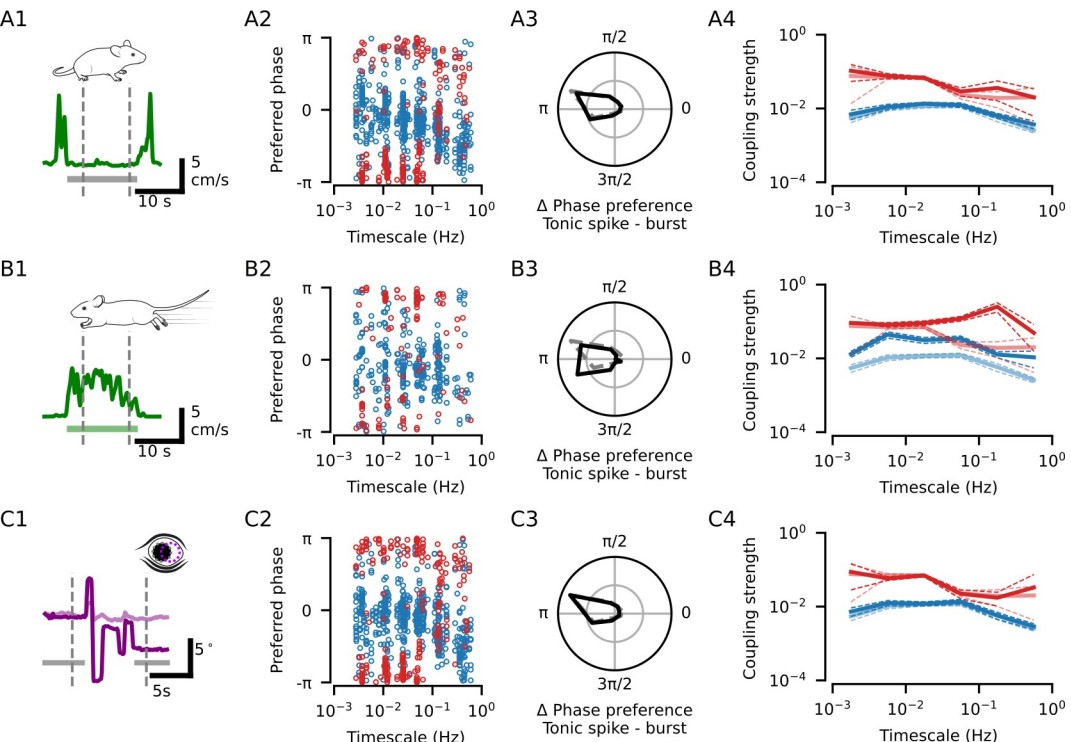

**Fig 4. The coupling between dLGN spiking and pupil dynamics is not driven by overt behaviors. (A)** Locomotion speed for an example period of quiescence (gray bar: period of quiescence). Dashed lines: period during which spikes were taken for the subsequent phase coupling analysis; note that the 4 s after the offset of the first locomotion bout and 2 s prior to the next locomotion bout were not included in this time window. **(A2)** Preferred coupling phase of tonic spikes (blue) and bursts (red) for all neuron-CPD pairs with significant coupling during quiescence. Significant tonic spike coupling was observed in 97.3% of neurons (143/147; mean = 4.2 CPDs per neuron) and burst coupling in 77.1% of neurons (84/109; mean = 2.3 CPDs per neuron). **(A3)** Distribution of the preferred phase differences between tonic spiking and bursting for neuron-CPD pairs with significant coupling of both spike types (mean = 2.9, $N$ = 221 neuron-CPD pairs; V-test for non-uniform distribution with a mean of $\pi$: V = 111.1, $p < 0.001$; grid lines indicate proportion of 0.25). **(A4)** Coupling strengths (median ± bootstrapped SE) measured during quiescence (bold lines) or across the whole recording (faded lines) for tonic spikes (blue; quiescence median = 0.0100, $N$ = 591 neuron-CPD pairs vs. overall median = 0.0081, $N$ = 681 neuron-CPD pairs; Mann–Whitney U test: U = $1.9 \times 10^5$, $p = 2.6 \times 10^{-2}$) and bursts (red; quiescence median = 0.0564, $N$ = 256 vs. overall median = 0.0528, $N$ = 320 neuron-CPD pairs; Mann–Whitney U test: U = $3.9 \times 10^4$, $p = 0.28$). **(B1)** Locomotion speed for an example bout (green bar: period of locomotion). Dashed lines: period during which spikes were taken for the subsequent phase coupling analysis; note that the first and final 1 s of the bout were not included in this time window. **(B2–5)** Same as (A2–5) but for phase coupling measured during locomotion bouts. Significant tonic spike coupling was observed in 85.3% of neurons (122/143; mean = 2.5 CPDs per neuron; median coupling strength during locomotion = 0.0254, $N$ = 359 neuron-CPD pairs; Mann–Whitney U test: U = $7.2 \times 10^4$, $p = 2.6 \times 10^{-28}$) and burst coupling in 77.9% of neurons (60/77; mean = 1.7 CPDs per neuron; median coupling strength during locomotion = 0.0927, $N$ = 133 neuron-CPD pairs; Mann–Whitney U test: U = $1.3 \times 10^4$, $p = 6.2 \times 10^{-10}$). Tonic spiking and bursting in neuron-CPD pairs with significant coupling of both spike types tended to occur at opposing phases (mean tonic spike—burst phase difference = 3.1, $N$ = 87 neuron-CPD pairs; V-test for non-uniform distribution with a mean of $\pi$: V = 56.7, $p < 0.001$). **(C1)** Eye position for an example recording period with saccadic eye movements (dark purple: azimuth; light purple: elevation; gray bars: the time windows without saccades). Dashed lines: period during which spikes were taken for the subsequent phase coupling analyses, note that the −2 to 2 s of activity surrounding each saccade was excluded. **(C2–5)** Same as (A2–5) but for phase coupling measured during periods with no saccadic eye movements. Significant tonic spike coupling was observed in 97.4% of neurons (151/155; mean = 4.1 CPDs per neuron; median coupling strength without saccades = 0.0090, $N$ = 634 neuron-CPD pairs; Mann–Whitney U test: U = $2.1 \times 10^5$, $p = 0.11$) and burst coupling in 81.7% of neurons (94/115; mean = 2.5 CPDs per neuron; median coupling strength without saccades = 0.0545, $N$ = 284 neuron-CPD pairs; Mann–Whitney U test: U = $4.5 \times 10^4$, $p = 0.99$). Tonic spiking and bursting in neuron-CPD pairs with significant coupling of both spike types tended to occur at opposing phases (mean tonic spike—burst phase difference = 2.9, $N$ = 203 neuron-CPD pairs; V-test for non-uniform distribution with a mean of $\pi$: V = 131.5, $p < 0.001$). CPD, components of pupil dynamics; dLGN, dorsal lateral geniculate nucleus.

Apart from locomotion, other behaviors, such as eye movements, may also be associated with arousal [33] and can induce changes in spiking activity. Indeed, upon visual inspection of eye position data, we noticed that saccades tended to occur during the dilation phase of large, slow pupil fluctuations, but also appeared to be related to smaller, faster fluctuations (S5A Fig), resembling the relationship between tonic spiking and pupil dynamics. To explore this further, we performed the same phase coupling analysis for saccades as we did for spiking and found that saccades had an almost identical phase coupling profile to that of tonic spikes: saccades predominantly occurred during the dilating phases of multiple temporal scales of pupil dynamics (S5B Fig; significant coupling of saccades to at least 1 CPD in 15/15 recording sessions; mean = 3.7 CPDs per session). To further investigate this relationship, we next asked if saccades could drive changes in spiking activity. Consistent with previous findings in primates (reviewed in [34]), we observed that bursting and tonic spiking in dLGN neurons was modulated in a short timescale window surrounding saccades. In a 4-s window surrounding saccades, tonic spiking increased, and bursting decreased, with the exception of a brief increase at the time of the saccade (S5C$_1$ Fig; $N$ = 118/121 neurons had significant peri-saccadic modulation). With regards to tonic spiking, neurons displayed diverse peri-saccadic activity patterns, among which we identified at least 2 distinct response types (S5C$_2$ Fig). Despite this diversity in profiles, the overall effect of this modulation was a consistent increase in tonic spiking in 89.0% of saccade-responsive neurons (S5C$_3$ Fig; modulation strength median = 0.20; Wilcoxon rank-sum test: W = $5.4 \times 10^2$, $p = 1.6 \times 10^{-15}$). In contrast, bursting activity tended to decrease (modulation strength median = −0.06; Wilcoxon rank-sum test: W = $1.4 \times 10^3$, $p = 6.2 \times 10^{-3}$). We conclude that saccades occur during dilating phases of pupil dynamics and have a marked impact on dLGN spiking activity.

We therefore hypothesized that the changes in spiking activity during the peri-saccadic period might contribute to the observed coupling between tonic spiking and pupil dynamics. We reasoned that, if the peri-saccadic modulation was driving the coupling between tonic spikes and pupil dynamics, then coupling strengths and saccadic modulation strengths should be correlated. Indeed, we found that saccadic modulation strengths could correlate with coupling strengths, but only for fast timescales of pupil dynamics, and only for tonic spiking (S5D Fig). To eliminate the effect of saccade-driven changes in spiking activity, similar to our approach for locomotion, we excluded spiking activity in a window from −2 s to 2 s surrounding saccades (Fig 4C$_1$) and repeated the phase coupling analysis. We found that the main characteristics of the coupling between spiking activity and pupil dynamics were preserved across temporal scales, even without peri-saccadic activity. Significant coupling for tonic spiking was observed in 97.4% of neurons (151/155; mean = 4.1 CPDs per neuron) and 81.7% of neurons for bursting (94/115; mean = 2.5 CPDs per neuron; Fig 4C$_2$). Tonic spiking and bursting retained their anti-phase relationship within neurons (mean tonic spike = burst phase difference = 2.9; V-test for non-uniform distribution with a mean of $\pi$: V = 144.0, $p = 0.0$; Fig 4C$_3$), and coupling strengths were similar (median tonic spike coupling strength without saccades = 0.0090, $N$ = 634 neuron-CPD pairs; Mann–Whitney U test: U = $2.1 \times 10^5$, $p = 0.11$; median burst coupling strength without saccades = 0.0545, $N$ = 284 neuron-CPD pairs; Mann–Whitney U test: U = $4.5 \times 10^4$, $p = 0.99$; Fig 4C$_4$). We therefore conclude that, although behaviors like locomotion and eye movements are reflected in some components of pupil size dynamics, and can induce changes in spiking, the coupling between spiking activity and pupil dynamics is not dominated by modulations related to these overt behaviors.

So far, we reported a multi-scale coupling of dLGN activity to pupil dynamics in the absence of a patterned visual stimulus, but how stable is this coupling in the presence of a rich visual stimulus? At least 2 factors could potentially disrupt this coupling. Firstly, naturalistic stimuli have been shown to elicit repeated patterns of both tonic and burst firing in the dLGN

 

[32,35], which may dominate internally driven activity fluctuations. Secondly, luminance changes or other salient features of the stimulus could induce pupil size changes that might interfere with those driven by internal state fluctuations. To investigate dLGN spike coupling to pupil dynamics during stimulus viewing, we presented 5-s long naturalistic movie clips to mice while recording pupil size and dLGN activity (Fig 5; data from [32]). We observed that many neurons in the dLGN responded to the movies with repeated patterns of activity, as exemplified by the neuron shown in Fig 5A$_1$. However, the average response to the stimulus (Fig 5A$_1$, bottom) often appeared weaker than the trial-to-trial variability in the time-averaged mean firing rates (Fig 5A$_1$, right). We quantified this difference using the "response variability ratio" (RVR; see Materials and methods) and discovered that 100% of neurons had an RVR below 1 ($N = 64$ neurons; Fig 5A$_2$). In fact, for 90.6% of neurons, the variability was more than twice as strong as the signal (RVR $< 0.5$; Fig 5A$_2$), indicating that dLGN neurons showed considerable variability in their firing rates that could not be accounted for by the visual stimulus. Similarly, when we repeated the same analysis for pupil size, we found that the movies did not systematically drive pupil size changes (Fig 5B$_1$), resulting in an overwhelming dominance of trial-to-trial variance in pupil size (Fig 5B$_2$). Thus, internally driven pupil dynamics remain largely uninterrupted by the naturalistic movie we presented. The analyses above suggest that factors unrelated to the stimulus, such as coupling to arousal-related variables, may be prevalent despite the spiking responses evoked by the movie stimulus.

To investigate the coupling of spiking activity to pupil dynamics during naturalistic stimulus viewing, we decomposed the pupil size signal as before and characterized the phase coupling of bursts and tonic spikes ($N = 9$ recording sessions in 6 mice). We found significant tonic spike coupling in 98.4% of neurons (62/63; mean = 4.3 CPDs per neuron) and burst coupling in 62.7% of neurons (32/51; mean = 1.3 CPDs per neuron; Fig 5C$_1$), with opposite phase preferences at slower CPDs (timescales $< 0.1$ Hz; Fig 5C$_2$). Although stimulus presentation appeared to disrupt the preferred phase relationship between bursts and tonic spikes for faster CPDs (timescales $> 0.1$ Hz; Fig 5C$_2$), the opposing-phase relationship was still present overall (mean tonic spike—burst phase difference = 2.85, $N = 56$ neuron-CPD pairs with significant coupling to both spike types; V-test for non-uniform distribution with a mean of $\pi$: V = 16.0, $p = 1.6 \times 10^{-3}$; Fig 5C$_3$). The disruption at faster timescales is likely due to the presence of stimulus-driven bursts [35,36], and also demonstrates that the opposing phase relationship is not guaranteed based on intrinsic biophysical properties of dLGN neurons, such as the burst generation mechanism. Coupling strengths tended to be lower for tonic spikes during movies (median = 0.0050, $N = 273$ neuron-CPD pairs) compared to gray screen conditions (median = 0.0081, $N = 681$ neuron-CPD pairs; Mann–Whitney U test: U = $1.1 \times 10^5$, $p = 1.1 \times 10^{-3}$; Fig 5C$_4$). In contrast, coupling strengths remained similar for bursts (movie median = 0.0332, $N = 68$ versus gray screen median = 0.0528, $N = 320$ neuron-CPD pairs; Mann–Whitney U test: U = $1.2 \times 10^4$, $p = 0.16$; Fig 5C$_4$).

To examine the implications of these findings for the encoding of arousal-related variables by dLGN spiking activity, we leveraged the repeating patterns of activity induced by the stimulus and next asked if pupil dynamics could be decoded from patterns of spiking activity. We first split each 5 s trial into 1 s segments, and, for each CPD, used the mean phase during each segment to assign one of 2 phase labels. We then used spiking activity from each individual neuron with significant phase coupling to decode the phase label. We found that the phase labels could be decoded above chance level for every timescale of pupil dynamics, both when using tonic spikes (S6A Fig) or bursts (S6B Fig). Together, these results show that dLGN neurons also represent multi-scale aspects of pupil dynamics during encoding of a naturalistic visual stimulus.

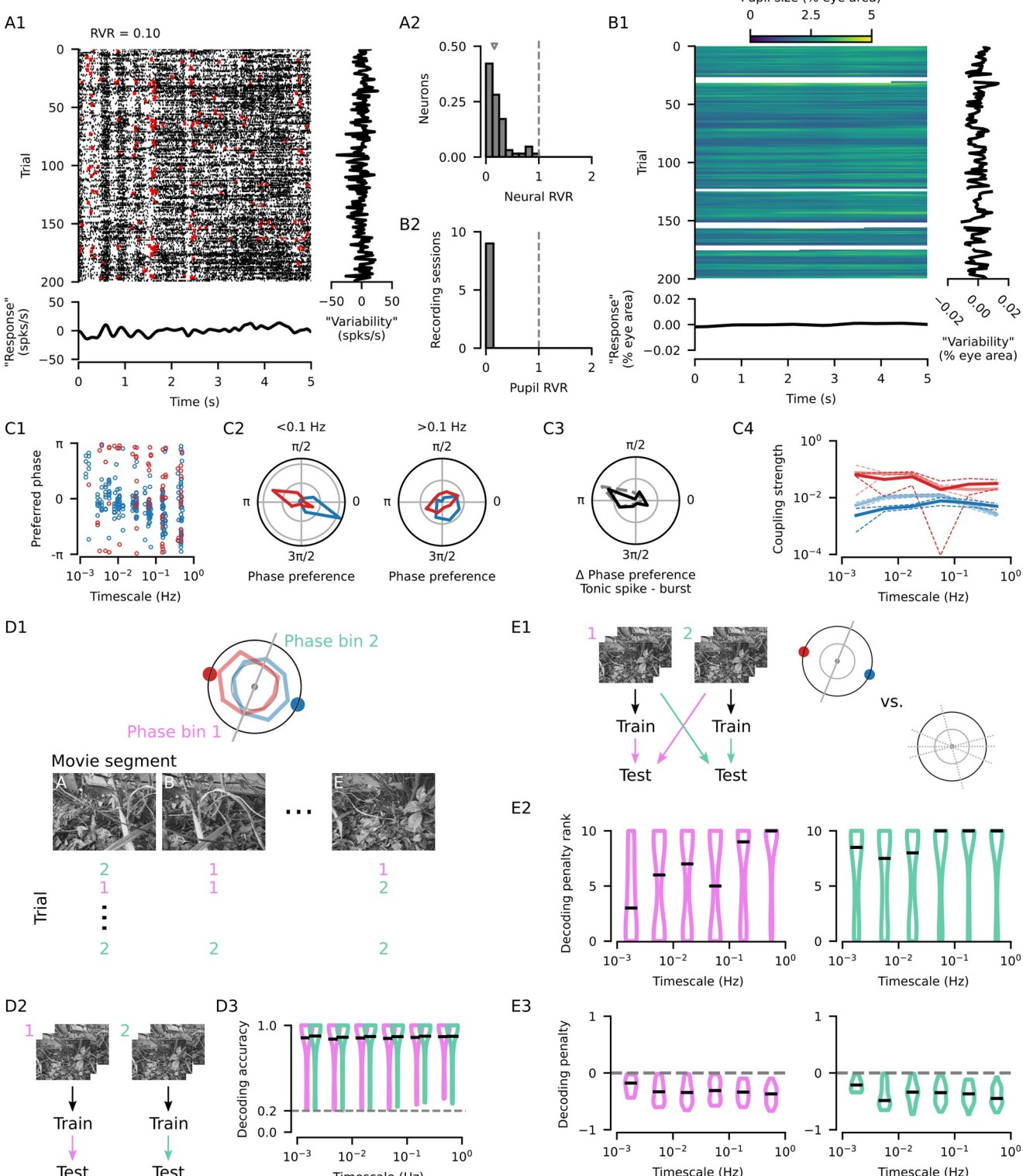

**Fig 5. Coupling of dLGN activity to pupil dynamics is related to differences in visual stimulus encoding. (A1)** Raster plot showing the responses of an example neuron to a movie (black: tonic spikes; red: burst spikes). Bottom: The mean response of the example neuron, centered on the mean across time. Right: The mean firing rate on each trial, centered on the mean across trials. Note that the "Response" and "Variability" axes have the same scale. **(A2)** The "response variability ratio" (RVR) compares the variance in the mean stimulus response to the mean variance for matching time points across trials (median = 0.16, N = 64 neurons; see Materials and methods). An RVR less than 1 indicates that the trial-to-trial changes in stimulus responses are larger

than the change induced by the stimulus. **(B1)** Trial-by-trial pupil size responses for the same experiment as in (A1). **(B2)** Distribution of pupil RVRs for all recording sessions with naturalistic movie stimulus presentation ($N$ = 9 recording sessions in 6 mice). An RVR of near zero indicates that stimulus induced fluctuations in pupil size were negligible compared to across-trial fluctuations. **(C1)** Preferred coupling phase of tonic spikes (blue) and bursts (red) for all neuron-CPD pairs with significant coupling during stimulus presentation. Significant tonic spike coupling was observed in 98.4% of neurons (62/63; mean = 4.3 CPDs per neuron) and burst coupling in 62.7% of neurons (32/51; mean = 1.3 CPDs per neuron). **(C2)** Preferred phase distribution for tonic spikes and bursts for slower timescales (left: timescale < 0.1 Hz; $N$ = 183 neuron-CPD pairs with tonic spike coupling; $N$ = 37 neuron-CPD pairs with burst coupling; grid lines indicate proportion of 0.25) and faster timescales (right: timescale > 0.1 Hz; $N$ = 90 neuron-CPD pairs with tonic spike coupling; $N$ = 31 neuron-CPD pairs with burst coupling). **(C3)** Distribution of the preferred phase differences between tonic spiking and bursting for neuron-CPD pairs with significant coupling of both spike types (mean = 2.85, $N$ = 56 neuron-CPD pairs, V-test for non-uniform distribution with a mean of $\pi$: V = 16.0, $p$ = 1.6 × 10$^{-3}$). **(C4)** Coupling strengths (median ± bootstrapped SE) measured during stimulus presentation (bold lines) or without patterned visual stimulation (faded lines, Fig 2E) for tonic spikes (movie median = 0.0050, $N$ = 273 neuron-CPD pairs vs. gray median = 0.0081, $N$ = 681 neuron-CPD pairs; Mann–Whitney U test: U = 1.1 × 10$^5$, $p$ = 1.1 × 10$^{-3}$) and bursts (movie median = 0.0332, $N$ = 68 vs. gray median = 0.0528, $N$ = 320 neuron-CPD pairs; Mann–Whitney U test: U = 1.2 × 10$^4$, $p$ = 0.16). **(D1)** Schematic representation of how the data was partitioned in (D) and (E). Each trial of naturalistic stimulus presentation was split into five 1-s segments and a decoder was trained to identify the stimulus segment using spiking activity. Two decoders were trained for each neuron-CPD pair, one for each of 2 CPD phase groupings. Turquoise: Decoders trained on segments during which the CPD was in the phase bin where tonic spiking preferentially occurred. Pink: Decoders trained on segments during which the CPD was in the opposing phase bin. **(D2)** Schematic of the movie segment decoding, initially performed by training and testing within data partitions. **(D3)** Decoding accuracy distributions for the 2 phase-groupings for each CPD timescale (black lines: median decoding accuracy; gray line: chance-level performance). **(E1)** Schematic of the movie segment decoding, now performed by training a decoder on one of the data partitions and testing it on data from the other. The decoding performance across CPD phases was compared to the scores from (D3) to yield a "decoding penalty." This penalty was compared to the penalty obtained by repeating the same procedure on 10 random partitions of the data segments. A decoding penalty was considered significant if it was greater than the penalties obtained from all 10 random partitions. **(E2)** Distribution of the ranks of decoding penalties obtained from partitions based on CPD phases among those obtained from 10 random partitions (black lines: median rank). High ranks indicate that the penalty for training and testing across data partitions based on CPD phases were greater than those obtained by random partitions. **(E3)** Distributions of decoding penalties, only for neuron-CPD pairs with significant penalties (black lines: median decoding penalty). The penalty for training and testing across CPD phases can be as high as 50% across most timescales, suggesting that all timescales of modulation contribute to substantial differences in how stimuli are encoded. CPD, components of pupil dynamics; dLGN, dorsal lateral geniculate nucleus; RVR, response variability ratio.

Given the different stimulus-response functions of bursts and tonic spikes [35–37], and changes in feature selectivity that can occur with arousal [38–40], we next asked whether the movie stimulus is differently represented by spiking activity during different phases of pupil dynamics. To this end, we used spiking activity to decode which of the five 1-s stimulus segments was presented. For each neuron-CPD pair, we sorted the stimulus segments into 2 groups according to the phase of the component at the time when the segment was on screen (Fig 5D$_1$). These 2 phase groupings were designed such that one phase bin was centered on the preferred phase of tonic spiking, while the other phase bin was centered 180° opposite. Critically, we used all spikes to decode the identity of the stimulus segment, regardless of whether they were tonic spikes or part of a burst, to simulate the perspective of a downstream neuron in the cortex that would be blind to this classification.

We observed that decoders trained and tested on data from the same phase group (Fig 5D$_2$) achieved nearly perfect performance (Fig 5D$_3$; phase bin 1 decoding accuracy median = 99.6%, phase bin 2 decoding accuracy median = 99.6%, $N$ = 399 neuron-CPD pairs), indicating that spiking activity during either phase was informative about the stimulus segment. However, when we tested the decoder trained using spiking activity from one phase group with activity occurring during the opposite phase grouping (Fig 5E$_1$), we found a marked decrease in decoding performance (Fig 5E$_3$; median penalty for train 1 → test 2 decoding = −33.6%, train 2 → test 1 = −34.5%). Critically, this decoding penalty was only considered significant if it was greater than the penalties obtained by performing the same cross-group decoding analysis on random partitions of the data (Fig 5E$_2$). The overall negative decoding penalty implies that encoding of the stimulus into firing rates differs between the phases of pupil dynamics. Overall, high within-state decoding accuracy and prominent decoding penalties across states suggest that the trial-to-trial variability in Fig 5A is not random but may stem from arousal-related changes in the stimulus-response properties of dLGN neurons. Importantly, these results were not limited to a specific temporal scale of pupil dynamics but were observed across

the entire range of timescales examined, indicating that multiple temporal scales of modulation affect stimulus encoding.

## Discussion

Our results establish that activity in dLGN, the primary visual thalamic nucleus, is coupled to distinct CPDs that occur over multiple timescales, ranging from several seconds to several minutes. Throughout these timescales, bursts and tonic spikes exhibited robust phase preferences, occurring during opposite phases of pupil dynamics across all timescales. Individual neurons were coupled to multiple timescales and were diverse in the timescale that drove the strongest modulation. The coupling between dLGN spiking and pupil dynamics extended beyond effects related to differences in pupil size per se and could not be attributed to transitions between quiescence and locomotion, or to saccadic eye movements. Furthermore, the coupling persisted even during viewing of a rich naturalistic movie, where we observed differences in how visual stimuli were encoded across phases of pupil dynamics. Together, our findings support the notion that arousal-related modulation of visual thalamus in the waking state results from a combination of changes in dLGN spiking activity linked to diverse temporal scales.

Our data-driven decomposition of the pupil signal moves beyond a binary classification of behavioral state and reveals CPDs which together predict dLGN activity modulations across multiple timescales. It is important to note that the components we recovered here do not necessarily map directly onto the time-courses of independent internal processes or behavioral sequences. Yet, our results support the idea that modulations of both the pupil signal and dLGN neural activity result from a combination of multiple intrinsic and/or extrinsic influences, possibly separated by the temporal scale over which they occur. Multi-scale influences have been described in other contexts, including the presence of diverse activity timescales within and between neurons [41–43], the action of several behavioral state-related neuromodulators with distinct timescales [24,44] or the largely non-overlapping activation dynamics of different ensembles of neurons within the same neuromodulatory system [45]. Furthermore, quantification of behavior in freely moving animals has revealed a multi-scale organization, where long-lasting "idle" states can be punctuated by rapid "active" behaviors (and vice-versa; [46]), mirroring the modulations we observed here, where slow and fast modulations were superimposed in dLGN activity. Finally, in the context of task performance, cognitive processes orchestrated by arousal-related neuromodulators can change on fast and slow timescales [21], such as quick reorienting to surprising stimuli or the choice of specific behavioral strategies over longer timescales. In humans, variations in stimulus detection ability have been linked to the phase of slow components (0.01 to 0.1 Hz) of EEG signals [47]. Thus, the multi-scale modulation we observed in the dLGN could reflect sensory processes adapting to the various timescales of behavioral organization of the organism.

While providing a quantitative account of dLGN modulations linked to multi-scale pupil dynamics, locomotion, and saccades, a limitation of our study is that it has not addressed these modulations on a mechanistic level. However, the relationship to pupil size strongly implicates the neuromodulators norepinepherine (NE) [20,24,48] and acetylcholine (ACh) [11,24] in the present findings. NE is provided to the thalamus by the locus coeruleus (LC) [49,50], while ACh in the rodent thalamus comes from several brainstem nuclei (PPN/LDT and PBG) [51–53] and the basal forebrain (BF) [51,54]. Consistent with a potential role for the LC-NE system in the present findings, stimulation of the LC in vivo has been shown to suppress bursting in the dLGN [55] and primary somatosensory thalamus (VPM) [56], while causing pupil dilation [48]. Additionally, NE levels in the mouse thalamus during NREM sleep have been found to

fluctuate with a frequency of 0.02 Hz and are linked to correlates of sensory arousability [57]. Given the prevalence of high power of CPDs around 0.02 Hz we observed, it also seems likely that the LC-NE system drives some of the coupling between dLGN spiking and pupil dynamics at such timescales during wakefulness. In addition, stimulation of the cholinergic PPT/LDT increases firing rates in the VPM [58], and switches dLGN neurons from burst to tonic mode [59], supporting the role of cholinergic nuclei in the modulations observed here. Although the activity of the PPN/LDT and PBG have not been directly linked to pupil size fluctuations, they are involved in coordinating locomotor patterns and eye movements [60,61], meaning they could specifically drive the subset of modulations we observed linked to these behaviors. It is important to note that neuromodulatory influences in the thalamocortical system extend beyond direct effects on thalamic neurons, and also involve the thalamic reticular nucleus [51,54,62–64], corticothalamic L6 neurons [65,66], and retinal boutons [40,67,68], all of which could contribute to additional arousal-related influences on thalamic neurons. Furthermore, although our analyses in darkness control for changes in retinal illumination linked to pupil size per se, there are known effects of pupil size on feed-forward signaling. Changes in retinal irradiation linked to pupil size can shift color selectivity in the visual system [39], and there is evidence for interactions between luminance and behavioral modulation [69] that warrants further exploration. Future studies directly tracking and manipulating the activity of these modulatory systems in the thalamus promise to yield further insight into the relationship between internal states and feed-forward sensory signals across diverse time scales.

The modulations in burst and tonic spiking we observed indicate that, throughout periods of wakefulness, and even during stimulus viewing, the dLGN is in constant alternation between 2 distinct information processing modes. One prominent view is that tonic spikes encode information about a stimulus linearly, while stimulus-driven bursts provide an all-or-none "wake-up call" to the cortex [14], by providing augmented retinogeniculate [37] and geniculocortical [18] communication, and improved stimulus detectability [70] (for related work in the somatosensory system, see e.g., [71,72]). Burst spikes, in comparison with tonic spikes, also have different feature selectivity: they track low frequency stimulus content [9,73,74] with an earlier response phase [32,36,75], integrate input over longer timescales with biphasic response kernels [35,36,76–78], and prefer smaller stimuli [79] with a stronger suppressive surround [36]. Such differences likely contribute to the decoding penalty we observed when training and testing decoders on spiking activity from opposing phases of the modulations we observed. However, it is also possible that neuromodulatory mechanisms change the stimulus response properties of dLGN neurons, without eliciting a switch in firing mode from tonic spiking to bursting. It remains to be seen whether spatiotemporal receptive fields change across phases of pupil dynamics, and whether these changes can be attributed to membrane hyperpolarization and bursting.

Our results focused on the commonalities of multi-scale modulation across the recorded population of dLGN neurons. However, we also observed a substantial diversity across neurons in the strength of coupling within a timescale, the timescale with strongest coupling, and the degree to which stimulus decoding was influenced. The mouse dLGN consists of various cell types [80] and functional subtypes [81,82], some of which are known to be differentially affected by arousal-linked variables [38,68]. Moving forward, considering the specific functional role of a given neuron, along with its temporal structure of modulation, will lead to a better understanding of dynamic stimulus processing in the visual system.

In this study, revealing the multi-scale nature of arousal-related activity modulations in dLGN was possible by applying a decomposition approach to the pupil signal. This approach builds on previous studies using raw pupil size as a marker for internal arousal processes in sensory circuits [7,40,66,83–87] or the pupil's dynamics within a restricted frequency range

[6,24,43]. Importantly, however, our analysis of pupil components and phases in relation to spiking activity is not intended to undermine the usefulness of pupil size itself, which remains an accessible signal that offers a relevant index into internal state. Indeed, our analyses show that modulations indexed by pupil size may be viewed as a specific case of phase coupling to pupil dynamics: coupling to high amplitude peaks or troughs of a component. We therefore view these approaches as complimentary and considering both allows for a more nuanced and comprehensive account of the modulations of sensory processing.

While our study has expanded the use of the pupil signal to reveal arousal-related modulations to multiple temporal scales, recent work has emphasized that a larger array of behaviors may also help to better explain activity in sensory systems [19,88]. Individual behaviors, such as locomotion onset [8,12], eye movements [89], head movements [69], or particular postures in freely moving conditions [90], modulate activity in the rodent early visual system. In the present study, locomotion and eye movements were shown to relate to a subset of pupil-linked modulations. However, given that a larger set of behavioral components has been linked to activity modulation in the visual cortex, this subset may expand as more behaviors are measured. These behavior-related modulations often occur in time-windows of seconds or less, which may be explained by the need to account for their immediate influence on sensory inputs [34,91]. Our present work shows that sensory neurons are modulated not only at these faster timescales, but also at longer timescales. In the future, a more complete characterization of the organization of spontaneous behaviors across temporal scales [46] promises to advance our understanding of multi-scale modulations of sensory processing by behavioral states.

## Materials and methods

### Ethics statement

All procedures complied with the European Communities Council Directive 2010/63/EU and the German Law for Protection of Animals, and were approved by local authorities (Regierung von Oberbayern, license #: ROB-55.2-2532.Vet_02-17-40; Regierungspräsidium Tübingen, license #: CIN 4/12), following appropriate ethics review.

### Surgical procedure

Experiments were carried out in 6 adult transgenic PV-Cre mice (median age at first recording session: 23.4 weeks; B6;129P2-Pvalb$^{tm1(cre)Arbr}$/J; Jackson Laboratory), 4 adult Ntsr1-Cre mice (median age: 24.7 weeks; B6.FVB(Cg)- Tg(Ntsr1-cre)GN220Gsat/Mmcd; MMRRC), and 1 wild-type BL6 mouse (age: 33.9 weeks), of either sex. Transgenic mice were used, as these mice were also included in another study [32] which required selective viral expression of ChR2 in area V1. For the present study, data mostly came from experiments where no optogenetic manipulation was present, with the exception of the data in Figs 5 and S6. Here, only trials without optogenetic stimulation were considered for the analyses.

The majority of experiments were performed under Licence ROB-55.2-2532.Vet_02-17-40. Thirty minutes prior to surgical procedures, mice were injected with an analgesic (Metamizole, 200 mg/kg, sc, MSD Animal Health, Brussels, Belgium). To induce anesthesia, animals were placed in an induction chamber and exposed to isoflurane (5% in oxygen, CP-Pharma, Burgdorf, Germany). After induction of anesthesia, mice were fixated in a stereotaxic frame (Drill & Microinjection Robot, Neurostar, Tuebingen, Germany). At the beginning of the surgical procedure, an additional analgesic was administered (Buprenorphine, 0.1 mg/kg, sc, Bayer, Leverkusen, Germany) and the isoflurane level was lowered (0.5% to 2% in oxygen), such that a stable level of anesthesia could be achieved as judged by the absence of a pedal reflex. Throughout the procedure, the eyes were covered with an eye ointment (Bepanthen, Bayer,

Leverkusen, Germany) and a closed loop temperature control system (ATC 1000, WPI Germany, Berlin, Germany) ensured that the animal's body temperature was maintained at 37°C. The animal's head was shaved and thoroughly disinfected using iodine solution (Braun, Melsungen, Germany). Before performing a scalp incision along the midline, a local analgesic was delivered (Lidocaine hydrochloride, sc, bela-pharm, Vechta, Germany). The skin covering the skull was partially removed and cleaned from tissue residues with a drop of $H_2O_2$ (3%, AppliChem, Darmstadt, Germany). Using 4 reference points (bregma, lambda, and 2 points 2 mm to the left and to the right of the midline respectively), the animal's head was positioned into a skull-flat configuration. The exposed skull was covered with OptiBond FL primer and adhesive (Kerr dental, Rastatt, Germany) omitting 3 locations: V1 (AP: −2.8 mm, ML: −2.5 mm), dLGN (AP: −2.3 mm, ML: −2 mm), and a position roughly 1.5 mm anterior and 1 mm to the right of bregma, designated for a miniature reference screw (00–96 X 1/16 stainless steel screws, Bilaney) soldered to a custom-made connector pin. Unrelated to the purpose of this study, 2 μl of the adeno-associated viral vector rAAV9/1.EF1a.DIO.hChR2(H134R)-eYFP.WPRE.hGH (Addgene, #20298-AAV9) was dyed with 0.3 μl fast green (Sigma-Aldrich, St. Louis, United States of America), and after performing a small craniotomy over V1, a total of approximately 0.5 μl of this mixture was injected across the entire depth of cortex (0.05 μl injected every 100 μm, starting at 1,000 μm and ending at 100 μm below the brain surface), using a glass pipette mounted on a Hamilton syringe (SYR 10 μl 1701 RN no NDL, Hamilton, Bonaduz, Switzerland). A custom-made lightweight stainless steel head bar was positioned over the posterior part of the skull such that the round opening contained in the bar was centered on V1/dLGN and attached with dental cement (Ivoclar Vivadent, Ellwangen, Germany) to the primer/adhesive. The opening was later filled with the silicone elastomer sealant Kwik-Cast (WPI Germany, Berlin, Germany). At the end of the procedure, an antibiotic ointment (Imax, Merz Pharmaceuticals, Frankfurt, Germany) was applied to the edges of the wound and a long-term analgesic (Meloxicam, 2 mg/kg, sc, Böhringer Ingelheim, Ingelheim, Germany) was administered and continued to be administered for 3 consecutive days. For at least 5 days post-surgery, the animal's health status was assessed via a score sheet.

One mouse was treated in accordance with Licence CIN 4/12, in which general surgical procedures were identical to the foregoing with the following exceptions. After induction of anesthesia, mice were additionally injected with atropine (atropine sulfate, 0.3 mg/kg, s.c.; Braun). The head post consisted of a small S-shaped piece of aluminum, which was cemented to the skull between lambda and bregma and to the right of the midline. Posterior to the head post, overlying the cerebellum, 2 miniature screws serving as ground and reference were implanted. At the end of the procedure, antibiotics (Baytril, 5 mg/kg, s.c.; Bayer) and a long-term analgesic (Carprofen, 5 mg/kg, s.c.; Rimadyl, Zoetis) were administered and were given for 3 days after surgery.

After at least 1 week of recovery, animals were gradually habituated to the experimental setup by first handling them and then simulating the experimental procedure. To allow for virus expression, neural recordings started no sooner than 3 weeks after injection. On the day prior to the first day of recording, mice were fully anesthetized using the same procedures as described for the initial surgery, and a craniotomy (ca. 1.5 mm$^2$) was performed over dLGN and V1 and re-sealed with Kwik-Cast (WPI Germany, Berlin, Germany). As long as the animals did not show signs of discomfort, the long-term analgesic Metacam was administered only once at the end of surgery, to avoid any confounding effect on experimental results. Recordings were performed daily and continued for as long as the quality of the electrophysiological signals remained high.

## Electrophysiological recordings

Mice were head-fixed on a styrofoam ball and allowed to run freely. Extracellular signals were recorded at 30 kHz (Blackrock Microsystems). For each recording session, the silicon plug sealing the craniotomy was removed. To record from dLGN, a 32-channel linear silicon probe (Neuronexus A1x32Edge-5mm-20-177-A32, Ann Arbor, USA) was lowered to a depth of approximately 2,700–3,700 μm below the brain surface. We judged recording sites to be located in dLGN based on the characteristic progression of RFs from upper to lower visual field along the electrode shank [81] and the presence of responses strongly modulated at the temporal frequency of the drifting gratings (F1 response). For post hoc histological reconstruction of the recording site, the electrode was stained with DiI (Invitrogen, Carlsbad, USA) for one of the final recording sessions.

For the purposes of a different study [32], during recordings involving naturalistic movie stimulation (Figs 5 and S6), V1 was optogenetically stimulated using 470 nm light on half of the trials, which were randomly interleaved with control trials. Here, only electrophysiological data from trials without optogenetic stimulation were considered.

## Histology

After the final recording session under Licence ROB-55.2-2532.Vet_02-17-40, mice were first administered an analgesic (Metamizole, 200 mg/kg, sc, MSD Animal Health, Brussels, Belgium) and following a 30 min latency period were transcardially perfused under deep anesthesia using a cocktail of Medetomidin (Domitor, 0.5 mg/kg, Vetoquinol, Ismaning, Germany), Midazolam (Climasol, 5 mg/kg, Ratiopharm, Ulm, Germany), and Fentanyl (Fentadon, 0.05 mg/kg, Dechra Veterinary Products Deutschland, Aulendorf, Germany). Perfusion was first done with Ringer's lactate solution followed by 4% paraformaldehyde (PFA) in 0.2 M sodium phosphate buffer (PBS).

To verify recording site and virus expression, we performed histological analyses. Brains were removed, postfixed in PFA for 24 h, and then rinsed with and stored in PBS at 4˚C. Slices (40 μm) were cut using a vibrotome (Leica VT1200 S, Leica, Wetzlar, Germany), mounted on glass slides with Vectashield DAPI (Vector Laboratories, Burlingame, USA), and coverslipped. A fluorescent microscope (BX61, Olympus, Tokyo, Japan) was used to inspect slices for the presence of yellow fluorescent protein (eYFP) and DiI. Recorded images were processed using FIJI [92,93].

For experiments under Licence CIN 4/12, general histological procedures were identical to those described above, except that mice were injected with sodium pentobarbital (Narcoren, = 200 mg/kg intraperitoneally; Böhringer Ingelheim) before perfusion. Coronal brain slices (50 μm) were obtained by using a vibratome (Microm HM 650V, Thermo Fisher Scientific) and inspected with a Zeiss Imager.Z1m fluorescent microscope (Zeiss).

## Visual stimulation

Visual stimulation was presented using custom written software (EXPO, https://sites.google.com/a/nyu.edu/expo/home) on a liquid crystal display (LCD) monitor (Samsung SyncMaster 2233RZ, 47×29 cm, 1680×1050 resolution at 60 Hz, mean luminance 50 cd/m$^2$) positioned at a distance of 25 cm from the animal's right eye. The data presented in all figures, with the exception of Figs 5 and S6, was recorded while animals were viewing either a static gray screen ($N$ = 5 experiments) or a sparse noise stimulus ($N$ = 10 experiments). The sparse noise stimulus consisted of a non-overlapping white and black square, simultaneously flashed for 200 ms on a square grid spanning 60 deg, while individual squares spanned 5 deg. Data was collected in darkness (S1 and S2 Figs) by switching off the display monitor and blocking smaller light

sources produced by the recording equipment. The data presented in Figs 5 and S6 were recorded while the animals were repeatedly presented with 5 s naturalistic movie clips (as described in [32]).

## Behavioral tracking

During electrophysiological recording, head-fixed mice were allowed to run on an air cushioned Styrofoam ball. Ball movements were recorded by 2 optical computer mice which interfaced with a microcontroller (Arduino Duemilanove) and sampled ball movements at 90 Hz. Locomotor activity was quantified by computing the Euclidean norm of 2 perpendicular components of ball velocity (pitch and roll), and herein referred to as locomotion speed. Locomotion bouts were defined as periods of time where the ball speed exceeded 1 cm/s for at least 2 s, with a break of no more than 2 s, and during which the locomotion speed exceeded the threshold for at least half of the bout duration. Quiescence was defined as any period of time outside of a locomotion bout.

To track pupil size, the stimulus-viewing eye was illuminated with and infrared LED light (850 nm), and the eye was filmed with a camera (Guppy AVT camera; frame rate 50 Hz, Allied Vision, Exton, USA) equipped with a zoom lens (Navitar Zoom 6000). Pupil size was extracted from the videos using a custom, semi-automated algorithm. Each video frame was equalized using an adaptive bi-histogram equalization procedure, and then smoothed using median and bilateral filters. The center of the pupil was initially estimated by taking the darkest point in a convolution of the filtered image with a black square. Next, the peaks of the image gradient along lines extending radially from the center point were used to define the pupil contour. Lastly, an ellipse was fit to the contour, and the area of the ellipse was taken as pupil size, and the center of the ellipse was taken as the pupil position. Frames in which the eye was obscured, occurring during eye closure or grooming, were detected by applying a threshold to the mean pixel-wise difference between each frame and a reference frame compiled by taking the median of several manually selected frames during which the eye was open. Data points with eye closure, as well as the 4 points immediately adjacent, were excluded. Because the pupil size and eyelid closure are correlated, many periods when the pupil was at its smallest could not be analyzed, including some periods of very low arousal and sleep-related states. Adjustable parameters in the above algorithm were set manually for each experiment. After ellipse fitting, data points at which the fitted pupil position, size, eccentricity, or rate of change, were outside of a plausible range were removed from consideration. Unreliable segments, occurring due to eye-closure, grooming, or unstable ellipse fitting, were automatically removed according to a priori criteria. Cubic splines were used to interpolate over gaps of <5 s, and the remaining segments of continuous data were smoothed with a Gaussian kernel ($\sigma$ = 250 ms). From each recording session, we only included data from the single longest continuous segment of reliable pupil data (median = 1,159 s, min = 596 s, max = 2,507 s). Pupil size is reported as a percentage of the total area of the exposed eye visible in the recording, as we found this to be a more stable normalization metric across recordings than the mean or maximum pupil size.

## Pupil size signal decomposition

The pupil size signal was decomposed into intrinsic mode functions (herein referred to as CPDs) by EMD (https://emd.readthedocs.io). In contrast to the original broad-band pupil size signal, the CPDs are locally narrow-band and thus amenable to Hilbert spectral analysis [29]. In contrast to the Fourier or wavelet transforms, EMD extracts these components without predefined filters and can capture nonstationarities present in biological signals with individual frequency- and amplitude-modulated components [29]. To minimize edge effects during

subsequent spectral analysis, each CPD was extrapolated from the beginning and end by mirroring the 3 closest peaks/troughs across the signal edge and interpolating between the mirrored extrema with third-order Bernstein polynomials restricted by the gradient at the signal edge. CPDs with fewer than 3 extrema were extended using the first derivative at the signal edges. The Hilbert transform was applied to the extended CPDs to obtain the complex-valued analytic signal of each CPD, from which the instantaneous phase and amplitude at each time point were computed as the angle and length of the signal vector, respectively. Instantaneous frequency was given by the time derivative of the unwrapped phase. Only time points corresponding to the un-extrapolated signal were used in further analysis. Pupil size power spectral density was compiled by binning the instantaneous frequency and collecting power (squared amplitude) for each frequency bin across CPDs and time (referred to as the "marginal spectrum" in [29]). This generally corresponded well with the PSD obtained via Fourier transform (median Pearson's R = 0.71, $N$ = 15 recordings). Each CPD was assigned a "characteristic timescale" by taking the mean frequency across all time points weighted by the amplitudes. The relative power of each CPD was assigned by taking the power (squared amplitude density of each CPD and expressing it as a fraction of the sum of the power densities from all CPDs extracted from the recording segment). At this point, some of the lower frequency CPDs were eliminated from further consideration if they did not complete at least 4 cycles within the recording segment to ensure sufficient sampling of each phase.

The relationship between pairs of CPDs from the same recording segment was assessed using a permutation test designed to find periods of phase coupling between CPDs. The CPDs resulting from EMD, although broadly different in frequency content, are not guaranteed to have independent phase progressions. Therefore, the same aspect of pupil dynamics can end up being captured by multiple CPDs. To address this problem, we reasoned that, if this mixing occurred, it would be reflected in transient periods of phase alignment between 2 CPDs. This alignment would appear as peaks in the joint phase distribution of 2 CPDs, causing the distribution to deviate from uniformity. However, because the phase distribution of each CPD itself is not uniform, for each pair of simultaneously recorded CPDs we simulated the expected joint distribution in the absence of phase coupling by shuffling the cycle order of each CPD independently 1,000 times. The real and simulated joint distributions were then compared to a uniform distribution using the Kullbach–Leibler divergence (Δ KLD, S1D Fig). The CPDs were considered significantly coupled if the Δ KLD of the true joint distribution exceeded the 95th percentile of the Δ KLDs computed from the simulated distributions ($p$-value $<$ = 0.05; S1D1 Fig, magenta outlines; S1D2 Fig, right). The specific combination of phases driving the coupling was determined by asking which points in the joint phase space exceeded the 95th percentile of the simulated distributions at the same point (S1D1 Fig, white outlines). These phase combinations were only considered significant if the Δ KLD also indicated overall coupling between the 2 CPDs. For the analysis in Figs 2F and S3B, the periods of time during which coupled CPDs passed through these phase combinations were excluded.

## Spike sorting

For recordings under protocol ROB-55.2-2532.Vet_02-17-40, the spiking activity of isolated single units was initially extracted from extracellular recordings using the Kilosort spike-sorting toolbox [94]. The resulting spike clusters were subject to manual curation in Spyke [95], where spikes within a cluster were temporally aligned and plotted in a 3D space (multichannel PCA, ICA, and/or spike time). In this space, clusters could be merged to account for drift in spike shape over the recording session (for example, if the first 2 wave shape PCs changed smoothly as a function of spike time), or further split using a gradient ascent-based clustering

algorithm [96]. Clusters containing only spikes with no consistent and clearly discernible voltage deflection were eliminated from further consideration. Finally, cluster auto-correlograms were examined to ensure that a clear refractory period was present; while the presence of a refractory period was not an indicator of a well-isolated unit, the absence of a refractory period was taken as an indicator that the cluster might need to be further split or contained a high amount of noise. Remaining clusters were compared using empirical distance metrics to ensure that they were well separated.

For the few sessions recorded under protocol CIN 4/12, single neurons in our linear array recordings were isolated by grouping neighboring channels into 5 equally sized "virtual octrodes" (8 channels per group with 2-channel overlap for 32 channel probes). Using an automatic spike detection threshold [97] multiplied by a factor of 1.5, spikes were extracted from the high-pass-filtered continuous signal for each group separately. The first 3 principal components of each channel were used for semi-automatic isolation of single neurons with KlustaKwik [98], and the resulting clusters were manually refined with Klusters [99]. Only clusters whose auto-correlogram displayed a clear refractory period and whose mean voltage trace showed a characteristic spike waveshape were further considered. To avoid duplication of neurons extracted from linear probe recordings, we computed cross-correlograms (1-ms bins) between pairs of neurons from neighboring groups. Pairs for which the cross-correlogram's zero bin was 3 times larger than the mean of nonzero bins were considered to be in conflict and only one was kept.

Bursts of action potentials are associated with a slow, hyperpolarization de-inactivated $Ca^{2+}$ conductance present in thalamic neurons, which cannot be directly measured in extracellular recordings. However, studies combining intra- and extracellular recordings have established reliable empirical criteria for identification of these thalamic bursts [30], according to which $\geq 2$ spikes with a prior period without spiking of 100 ms and an ISI of $<4$ ms are part of a burst (Fig 2A). Spikes satisfying these criteria were categorized as burst spikes, whereas all other spikes were considered tonic spikes. Downstream analyses were performed separately on tonic spikes and burst events, for which all spikes in a burst were treated as a single event. Among our recorded dLGN units, the vast majority (89.5%) displayed bursting events, which and accounted for 3% of dLGN spikes (burst ratio: the proportion of all spikes in the recording that were part of a burst; $S2A_2$ Fig).

Short timescale serial dependence of burst events and tonic spiking was assessed by computing the auto-correlogram for each activity type in a −1 s to 1 s window. Taking the average auto-correlation over all neurons, it was determined by visual inspection that both event types had a primary peak lasting approximately 300 ms. In addition, we also note that the mean burst auto-correlation had a secondary peak, indicating rhythmicity in the ∼5 Hz range [13].

## Modulation of spiking by pupil size

To assess the modulation of spiking activity by arousal states, we began by collecting, for each neuron, spike counts in 250 ms bins. We then sorted these spike counts into 10 bins according to the min-max normalized pupil size. A neuron was considered to be significantly modulated by pupil size if a one-way ANOVA across the pupil size bins was significant, regardless of which pupil size bin had the highest firing rate.

## Phase coupling

To assess the modulation of spiking activity by arousal states, we developed a phase coupling analysis to relate events such as bursts and tonic spikes to the CPDs. We required that at least 8 bursts or tonic spikes occurred over the course of the recording segment for a neuron to be considered in this analysis. Considering tonic spikes and burst events from each neuron

separately, the Hilbert phase of each ongoing CPD was collected at times when bursts (the time of the first spike of the burst) and tonic spikes occurred. Although coupling might be directly assessed by compiling phase histograms and computing the circular mean of these phases, we adopted various bias-corrections resulting in a more stringent assessment of phase coupling.

Firstly, CPDs are not guaranteed to have a linear phase progression, which is beneficial in that they can capture naturalistic asymmetric wave shapes but presents a difficulty when assessing phase coupling as the underlying distribution of phases could itself be biased. These biases could lead one to falsely infer phase coupling simply because more recording time may be spent in a certain phase, and therefore more spikes attributed to this phase. We therefore converted phases to circular ranks, yielding a uniform underlying distribution to which bursts and tonic spikes could be related [100]. The angle of the circular mean rank was then converted back to a phase using the original distribution, and this value was reported as the "preferred phase." Next, as the circular mean resultant vector length is a biased statistic (its value is inflated for small number of observations), we opted for an unbiased measure of phase coupling strength to allow comparison of coupling strength across units with different firing rates, between tonic spikes and bursts, and between conditions where spikes were subsampled from the recording period. Rather than using mean resultant vector length ($R$), we used its squared value, which can be easily bias-corrected [101]. We thus quantified coupling strength as follows:

$$\hat{R}^2 = \frac{n}{n-1}\left(R^2 - \frac{1}{n}\right),$$

where $R$ is the mean resultant vector length computed using the circular phase ranks, and $n$ is the number of tonic spikes or bursts. Finally, we sought to assess the statistical significance of phase coupling with a measure that was insensitive to the short-term serial dependence of spike trains, which effectively reduces the degrees of freedom of the sample. We adopted the null hypothesis that, if there were no relationship between spiking activity and CPD phase, then the coupling strength of a neuron-CPD pair would be the same as that measured from a spike train with the same short-term structure, but with no relationship to the CPD. We tested coupling strengths against this null hypothesis by splitting burst and tonic spike trains into segments of 300 ms, which were shuffled in order to destroy the relationship between CPD phase and spiking activity, and then phase coupling to the CPD was assessed as described above. This procedure was repeated 1,000 times for each neuron-CPD pair to compile a null-distribution of coupling strengths. The true coupling strength was then compared to this distribution and assigned an exact $p$-value (precise to 3 decimal places) based on how many elements of the null set had a higher coupling strength value.

In general, we computed phase coupling using all recorded spikes. However, to account for the influence of spiking activity surrounding behaviors such as locomotion and eye movements (Figs 4, S4, and S5), we excluded spiking activity that occurred during these behaviors from consideration. To measure coupling during periods of quiescence, spikes that occurred during locomotion bouts were excluded, as well as spikes occurring 2 s before bout onset, and 4 s after bout offset. For coupling during locomotion, spikes occurring during quiescence were excluded, as well as spikes occurring during the first and last 2 s of the bout, meaning that bouts shorter than 4 s in length were not considered. To eliminate the influence of peri-saccadic activity, we excluded spiking activity occurring in a time window from 1 s before to 2 s after the saccade. Finally, because 2 simultaneously recorded CPDs could themselves have phase coupling (S1D Fig), we assessed the coupling of single neurons to multiple CPDs after

removing periods of time where significant coupling between a given CPD and any other CPD in the set was observed according to the statistical procedure detailed above (Figs 2F and S3B). In the case of the phase coupling comparison across data partitions (S3D Fig), for each neuron and CPD, the spiking from adjacent CPD cycles was placed into one of the 2 partitions. Coupling was assessed for each partition separately, then the difference in preferred phase across partitions was compared for each neuron-CPD pair. For coupling strength ranking across CPDs, the partition with data from odd CPD cycles was arbitrarily assigned to 1 group, and even cycles to the other group. Coupling strength was then ranked within groups and the top CPD was compared across groups for each neuron.

## Pupil size distribution matching

We observed that pupil size could co-vary with CPD phase (Fig 3A), thus presenting a confound for genuine phase coupling. To control for effects of pupil size per se from the measurement of CPD phase coupling, we adopted a histogram matching procedure [102,103] to minimize the differences in the distribution of pupil sizes across the phase bins of each CPD. For each CPD, the recording was split into segments based on 4 phase bins. For each visit to a phase bin, the mean pupil size during the visit was collected and compiled into a histogram (10 bins). From these histograms (1 for each phase bin), we assembled a "minimum common distribution" of pupil sizes by selecting the phase bin with the smallest number of entries for each pupil size. For each of these entries in the selected phase bin, we collected the entry from the other 3 phase bins with the closest pupil size, thus obtaining an equal number of samples from each phase bin, with pupil sizes matched as closely as possible. To assess the efficacy of this procedure, we computed the pair-wise difference in mean pupil size (this time collecting the full time-course of pupil size from each bin visit, rather than the mean) between each pair of phase bins, and took the maximum difference. We compared this difference before and after the subsampling procedure (Fig 3A). The phase coupling analysis in Fig 3 results from taking only spikes that occurred during the time periods subsampled with the matching procedure.

## Correlation analyses

Locomotion speed is known to be reflected in the pupil size signal (S1B4 Fig, [8]). To assess the potential contributions of locomotion speed to the phase coupling we observed, we performed correlation analyses relating each CPD to the animal's locomotion speed. Because serial dependence in signals can inflate correlation values and violates the independence assumption required to directly calculate a $p$-value, we adopted a permutation-based approach to assess the statistical significance of CPD-speed cross-correlations [104]. Each CPD was correlated with 1,000 locomotion speed traces collected from other experiments and compared the peak value cross-correlation to the distribution of nonsense correlations in order to obtain a $p$-value. To further reduce the detection of spurious correlations, we limited our search for the peak to lags in the range of $[−T, T]$, where T is the mean period of the CPD. For example, for a CPD with a mean period of 5 s, the search window would be restricted to lags of $[−10$ s, 10 s], and maximum value of the absolute cross-correlation in this window would be compared to the distribution of nonsense correlations. The cross-correlation was considered significant only if the maximum value within the prescribed range had a $p$-value $\leq 0.05$.

## Spiking responses

To compute spiking responses to behavioral events (locomotion onsets, locomotion offsets, saccades) and experimental events (stimulus onsets), we first estimated the instantaneous firing rate surrounding each event via kernel density estimation ([105], implemented in the

statistics.instantaneous_rate function from https://elephant.readthedocs.io). Instantaneous rates were estimated using a Gaussian kernel with 100 ms bandwidth and sampled with a resolution of 50 ms. Responses were considered significant if at any point the mean response was outside the [2.5, 97.5] percentile range of responses computed using a shuffled version of the spike train (300 ms bins, 1,000 shuffles). In the case of saccade responses (S5C Fig), K-means clustering (sklearn.clustering.KMeans, with k = 3) was performed after taking the top principal components (explaining 80% of the variance) of the mean responses of each neuron in a window of [−1 s, 1 s] surrounding the saccade. To quantify the trial-to-trial variability in stimulus responses (Fig 5A2 and 5B2), we constructed a "response variance ratio" (RVR) comparing the variance across time of the mean stimulus response to the mean across-trial variance for each time point.

$$SNR = \frac{Var_l(E_k(X))}{E_l(Var_k(X))},$$

where $X = [x_1, \ldots, x_k]^T \in R^{K,L}$ is the response matrix compiled from $K$ trials of duration $L$, and $E_d()$ and $Var_d()$ denote taking the mean or variance across the indicated dimension of the matrix.

## Decoding analyses

For each neuron-CPD combination, we trained support-vector classifiers (SVCs; sklearn.svm.SVC) to decode CPD phase and visual stimulus identity from spiking activity during naturalistic stimulus viewing (Figs 5 and S6). Spike times (tonic spike times, burst times, or all spikes) from each trial were converted into instantaneous rates (see above), and each 5 s trial was split into 1 s segments. To decode the CPD phase (S6 Fig), we assigned a label to each 1 s segment of activity by taking the circular mean phase of the CPD during the segment and placing this mean phase into one of 2 phase bins. We then trained an SVC (with a radial basis function kernel) to decode the phase label using only burst rates or only tonic spike rates occurring during the 1 s segment and assessed the decoding performance with 5-fold cross-validation. To decode the visual stimulus (Fig 5D), we assigned each 1 s segment with a label (e.g., 1 for the first second of the stimulus, 5 for the last second of the stimulus), and then split the segments into 2 groups according to the mean CPD phase during the segment. We used the instantaneous rates (estimates using all spikes together regardless of their categorization as tonic or burst) from one group to train an SVC (with a linear kernel) to perform "one-versus-rest" decoding of the stimulus labels and tested the decoder using the same data as well as the data from the other group. We took the difference in decoding scores between the 2 groups as the "decoding penalty." We compared the decoding penalty obtained when splitting the segments by CPD phase to the decoding penalties obtained from 10 random splits of the data as a control. The decoding penalty for the neuron-CPD pair was considered significant if it was greater than all 10 random splits. In both of the above decoding schemes, we determined the 2 CPD phase bins based on the preferred coupling of each neuron: one bin was centered around the preferred phase of tonic spiking of the neuron (preferred phase ± $\pi/2$), and the other bin was centered around the opposing phase (preferred phase - $\pi$ ± $\pi/2$).

## Supporting information

**S1 Fig. Characterizations of pupil and running behavior, relationship of spikes to pupil size during darkness, and independence of CPDs. (A1)** Pupil size distributions for all recording sessions. Pupil size is expressed as a fraction of the total exposed eye area (see Materials and methods). Data from the gray screen and sparse noise sessions are grouped together for

the analyses in Figs 1, 2, 4, S2, S3B, S3C and S4. **(A2)** Pupil size power spectral density (mean ± SEM, min-max normalized) for gray screen recordings ($N = 5$) and recordings in darkness ($N = 5$). **(B1)** Distribution of proportion of time spent in a locomotion bout (green, see Materials and methods) versus sitting (*gray*) for $N = 14$ sessions where locomotion speed was recorded (Horizontal bars: median proportion). **(B2)** Distribution of locomotion bout and inter-bout-interval lengths. **(B3)** Distribution of locomotion speed variance during locomotion bouts versus sitting. Whereas locomotion bouts were characterized by larger behavioral variability, the inter-bout-intervals had very low variability in locomotion speed. **(B4)** Pupil size (mean ± SEM, normalized to the pre-bout pupil area) surrounding locomotion bout onsets (left) and offsets (right). **(B5)** Pupil size power spectral density (mean ± SEM) for periods of locomotion (green) and quiescence (gray). **(C1)** Spike counts (min-max normalized) across pupil sizes (min-max normalized) for the dLGN neurons with significant modulation by pupil size (top, one-way ANOVA across 10 pupil size bins, $p \leq 0.05$) and without significant modulation (bottom) during recordings performed in darkness. Neurons are sorted by the location of the maximum firing rate (black dots). The majority of significantly modulated neurons (61.8%) had "non-monotonic" modulation profiles, with their maximum firing rates outside of the 90th percentile of pupil size (dashed white line). **(C2)** Same as (C1) but for bursts of spikes. Neurons are sorted by the location of the maximum burst rate (black dots). The majority of significantly modulate neurons had "monotonic" modulation profiles, with their maximum burst rates in the 10th percentile of pupil size. **(C3)** For each dLGN neuron in the light (Fig 1C1) and dark (S1C1 Fig), the spike count Fano factor (spike count variance/mean) in each pupil size bin was computed, and the mean across pupil size was taken. The Fano factor is >1 in both the light (median Fano factor = 1.95, Wilcoxon rank-sum test for mean Fano factor >1: $W = 1.2 \times 10^4$, $p = 5.2 \times 10^{-26}$, $N = 156$ neurons) and dark (median Fano factor = 1.38, mean Fano factor > 1: $W = 4.4 \times 10^3$, $p = 6.4 \times 10^{-16}$, $N = 94$ neurons). **(D1)** Joint phase distributions for the CPDs from the recording in Fig 1B, illustrating the results of our statistical test for phase coupling between CPDs (see Materials and methods). Magenta outlines: CPD pairs where the distribution as a whole showed that the CPDs were coupled ($p \leq 0.05$). White outlines: regions in the joint phase space where coupling might have occurred. **(D2)** Phase coupling z-scores quantifying the amount of coupling occurring between 2 simultaneously recorded CPDs from gray screen and sparse noise recordings. Top: CPD pairs without significant coupling (77.0%, 354/460 CPD pairs). Bottom: CPD pairs with significant phase coupling (23.0%, 106/460 CPD pairs). The majority of CPD pairs did not have coupling, suggesting that they represent independent aspects of pupil dynamics. Those that had coupling tended to be similar in temporal scale (i.e., were close to the diagonal).
(TIFF)

**S2 Fig. Firing rate distributions, auto- and cross-correlograms of tonic spiking and bursting. (A1)** Distribution of firing rates (black) for all neurons recorded in spontaneous or sparse noise sessions (mean = 4.49 spk/s, $N = 156$). Blue: The firing rate distribution considering only tonic spikes (mean = 4.11 spk/s), showing that overall firing rates were primarily determined by tonic spiking. Neurons with a firing rate <0.01 spk/s were excluded from all analyses. **(A2)** Distribution of burst ratios (number of spikes assigned to a burst/total number of spikes, median = 3.0%). Bursts were detected in 92.9% of neurons (145/156). **(B1)** Mean auto-correlogram of tonic spiking (mean ± SEM). The dashed line at 300 ms indicates the bin-width used to generate shuffled spike trains used to test phase coupling significance (see Materials and methods). **(B2)** Mean auto-correlogram of bursting (mean ± SEM). Note that the peaks are spaced apart by approximately 200 ms, indicating that approximately 5 Hz rhythmic bursting was present (see also Nestvogel and colleagues). **(C)** Mean burst cross-correlogram

(mean ± SEM) for all simultaneously recorded pairs of neurons ($N$ = 1,185 pairs). The peaks at zero and approximately 200 ms indicate that neurons tend to burst synchronously and rhythmically.
(TIFF)

**S3 Fig. Coupling of spikes to CPDs in darkness and detailed characterization of CPD preferences. (A**) Phase coupling analysis relating dLGN spiking to CPDs for recordings performed in darkness. **(A1)** Preferred coupling phase of tonic spikes (blue) and bursts (red) for all neuron-CPD pairs with significant coupling. Significant tonic spike coupling was observed in 96.8% of neurons (91/94; mean = 3.4 CPDs per neuron) and burst coupling in 64.8% of neurons (35/54; mean = 1.9 CPDs per neuron). **(A2)** Distribution of the preferred phase differences between tonic spiking and bursting for neuron-CPD pairs with significant coupling of both spike types (mean = 2.5; $N$ = 88 neuron-CPD pairs; V-test for non-uniformity and a mean of $\pi$: $V$ = 25.8, $p$ = 5.1×10$^{-5}$; grid lines indicate proportion of 0.25). While the mean phase difference is similar to Fig 2D (dashed gray line), we also note that in darkness a small proportion of neurons appear to have a spiking pattern where tonic spikes immediately follow bursts without a phase delay ($\Delta$ phase preference ~$2\pi$). **(A3)** Coupling strengths (solid lines: median, dashed lines: bootstrapped SE of the median) measured in darkness (bold lines) or in an illuminated environment (faded lines; Fig 2E) for tonic spiking (dark median = 0.0058, $N$ = 321 neuron-CPD pairs vs. illuminated median = 0.0081, $N$ = 682 neuron-CPD pairs; Mann–Whitney U test: $U$ = 1.2×10$^5$, $p$ = 2.9×10$^{-3}$) and bursting (dark median = 0.0386, $N$ = 100 neuron-CPD pairs vs. illuminated median = 0.0528, $N$ = 320 neuron-CPD pairs; Mann–Whitney U test: $U$ = 1.8×10$^4$, $p$ = 0.05). **(B)** Phase coupling analysis performed on the same recordings as in Fig 2, but excluding spikes that occurred during periods of phase coupling between CPDs (S1D Fig; see Materials and methods). **(B1)** Preferred coupling phase of tonic spikes (blue) and bursts (red) for all neuron-CPD pairs with significant coupling. Significant tonic spike coupling was observed in 98.6% of neurons (146/148; mean = 3.8 CPDs per neuron) and burst coupling in 74.2% of neurons (89/120; mean = 2.1 CPDs per neuron). **(B2)** Distribution of the preferred phase differences between tonic spiking and bursting for neuron-CPD pairs with significant coupling of both spike types (mean = 2.7; $N$ = 203 neuron-CPD pairs; V-test for non-uniformity and a mean of $\pi$: V = 109.1, $p$ < 0.001; grid lines indicate proportion of 0.25). **(B3)** Coupling strengths (median ± bootstrapped SE) measured after removal of periods of phase coupling between CPDs (bold lines) or over the whole recording (faded lines; Fig 2E) for tonic spiking (no coupling median = 0.0089, $N$ = 563 neuron-CPD pairs vs. overall median = 0.0081, $N$ = 682 neuron-CPD pairs; Mann–Whitney U test: $U$ = 1.9×10$^5$, $p$ = 0.34) and bursting (no coupling median = 0.0583, $N$ = 250 neuron-CPD pairs vs. overall median = 0.0528, $N$ = 320 neuron-CPD pairs; Mann–Whitney U test: $U$ = 3.9×10$^4$, $p$ = 0.56). **(C1)** For each recording session, the mean preferred timescale across neurons plotted against the frequency of behavioral state switches (1/mean duration of locomotion and quiescence periods). The correlation (Pearson's $R$ = 0.71, $p$ = 4.3×10$^{-3}$, $N$ = 14 recording sessions) indicates that part of variability in timescale preferences between recordings (Fig 3G) is related to differences in behavior. **(C2)** Proximity of the CPDs with the strongest and second strongest coupling for tonic spikes (blue) and bursts (red), a value of 1 indicates that the CPD with the second strongest coupling is adjacent in characteristic timescale to the strongest. These results suggest that although neurons may be modulated at multiple timescales, the range of temporal scales with strong modulation is limited. **(D)** Preferred timescale and phase tend to be stable across data sub-samples (blue: tonic spikes, red: bursts). The phase coupling of individual neurons was compared across each of 2 interleaved partitions of the data (see Materials and methods). The change in the CPD to which a neuron is most strongly coupled is shown in (D1) for

all neurons, 0 indicates that the strongest CPD remained unchanged, 1 indicates that the strongest CPD shifted to a CPD adjacent in characteristic timescale. The overwhelming majority of neurons either retain the same preferred CPD or switch to the next closest timescale. The change in preferred phase for each neuron-CPD pair is shown in (D2) for pairs in which the coupling was significant in both subsamples (colored distribution) and in which coupling did not reach significance in one or both of the subsamples (gray distributions). For both tonic spikes and bursts, the change in phase preference was 90˚ for >90% of neurons. **(E)** Proportion of neurons that shift (faded sections) or maintain (bold sections) their preferred coupling timescales after controlling for pupil size (Fig 3), split by the profile of modulation by pupil size (Monotonic: neurons with peak rate in the top decile for tonic spiking, or bottom decile for bursting vs. Other: all other neurons with significant modulation across pupil sizes vs. None: neurons with no modulation across pupil sizes; see Fig 1C). For tonic spikes (top), there was no significant difference between groups (mono = 37/63 vs. other = 43/65 vs. none = 6/11; Chi-squared test: $X^2 = 1.0$, $p = 0.60$). Meanwhile, neurons with monotonic burst modulation by pupil size (in contrast to tonic spiking, this class constituted the majority of neurons for bursting) were significantly more likely to switch preferred timescales (bottom; mono = 34/46 vs. other = 7/13 vs. none = 5/16; Chi-squared test: $X^2 = 9.5$, $p = 8.7 \times 10^{-3}$).
(TIFF)

**S4 Fig. Spiking activity and pupil dynamics are correlated with locomotion speed. (A1)** Firing rate responses to the offset of locomotion bouts (mean ± SEM, relative to the baseline from −5 to −3 s and min-max normalized for each neuron) for tonic spiking (blue; $N = 111/121$ neurons with significant tonic spiking modulation) and bursting (red; $N = 98/121$ neurons with significant bursting modulation). Vertical lines denote the period surrounding the offset of locomotion bouts during which the mean firing rate deviates from baseline (−1 s to 4 s), and therefore spiking activity during this transition period was excluded for the analysis in Fig 4A, in addition to all activity during locomotion bouts. **(A2)** Same as (A1) but for the onset of a locomotion bout ($N = 108/121$ neurons with significant tonic spiking modulation; $N = 96/121$ neurons with significant bursting modulation). Vertical lines denote the transition period (−2 s to 2 s) surrounding the onset of a bout from which spiking activity was excluded for the analysis in Fig 4B, in addition to all activity during quiescence. (A3) Median burst ratio (number of burst spikes/total number of spikes, error bars: bootstrapped SE of the median) across quiescence (gray, excluding 2 s prior to and following locomotion bouts) and locomotion (green, split for the first 2 s of bouts, the middle portion of bouts, and the final 2 s of bouts) showing burst spikes are less prevalent, but not absent, during locomotion (Friedman chi-squared test: $Q = 22.0$, $p = 6.7 \times 10^{-5}$). **(B)** Left: Mean cross-correlation between CPDs and locomotion speed, grouped by the timescale of the CPD (black dots: location of the peak correlation for each CPD with a significant correlation; gray dots: location of the peak correlation for each component without a significant correlation). Correlation significance was determined by comparing the maximum value of the cross-correlation to a null-distribution obtained by correlating the CPD with locomotion speed traces taken from different recording sessions (see Materials and methods). To eliminate nonsense correlations, we only considered the cross-correlation significant if the peak was found at lags shorter than the mean period of each CPD. Right: Proportion of CPDs in each timescale bin with a significant correlation to locomotion speed (significant correlation in 55.3% of CPDs, 52/94). **(C)** Same as (B), but only considering the CPD and locomotion speed within locomotion bouts (significant correlation in 64.9% of CPDs, 61/94).
(TIFF)

**S5 Fig. Saccades are linked to pupil dynamics and trigger changes in dLGN activity. (A)** Example eye position (top) and pupil size (bottom) traces. Detected saccades are marked by the dashed lines. **(B)** This coupling of saccades to pupil dynamics was verified using the same phase coupling analysis as in Fig 2, relating saccades to each of the CPD. Shown here are the preferred phases at which saccades occur, for CPDs across various timescales (significant saccade coupling to at least 1 CPD was observed in 15/15 recording sessions). Across all timescales, saccades tend to occur during pupil dilations, similar to tonic spikes (Fig 2C). **(C1)** Firing rate responses to saccades (mean ± SEM, relative to the baseline from −5 to −3 s and min-max normalized for each neuron) for tonic spiking (blue; $N = 118/121$ neurons with significant tonic spiking modulation) and bursting (red; $N = 88/121$ neurons with significant bursting modulation). Vertical lines denote the period surrounding saccades during which the mean firing rate deviates from baseline (−2 s to 2 s); therefore, spiking activity during this transition period was excluded for the analysis in Fig 4C. **(C2)** Neurons in the dLGN have diverse saccade-triggered tonic spiking responses. Clustering the normalized peri-saccadic responses (see Materials and methods) revealed at least 2 distinct response types, in addition to a minimally responsive/mixed cluster (left). The first responsive cluster (middle) had a transient increase in firing tightly locked to saccade onsets. The second responsive cluster (right) was characterized by gradually increased firing rates prior to saccade onset, with a brief suppression immediately following saccade onset, before returning to a sustained facilitation. **(C3)** Despite the diversity in responses, the peri-saccadic period (−2 to 2 s) was characterized by an overall increase in tonic spiking rates. The modulation was quantified by taking the area under the normalized saccadic response curve in a window spanning −2 to 2 s for tonic spiking (median = 0.2; Wilcoxon rank-sum test: $W = 5.4 \times 10^2$, $p = 1.6 \times 10^{-15}$, $N = 118$) and bursting (median = −0.06; Wilcoxon rank-sum test: $W = 1.4 \times 10^3$, $p = 6.2 \times 10^{-3}$, $N = 88$). **(D)** Pearson correlation between CPD coupling strengths and saccadic modulation strengths for each timescale (*p*-value denoted by *asterisks*, * for $p \leq 0.05$, ** for $p \leq 0.01$, *** for $p \leq 0.001$). The correlation was not significant for the majority of timescales, suggesting that the coupling between tonic spiking and pupil dynamics was independent from the rapid peri-saccadic changes in firing rate observed in (C).
(TIFF)

**S6 Fig. CPD phase can be decoded from spiking activity during stimulus viewing. (A)** Distributions of CPD phase decoding accuracy using tonic spiking ($N = 273$ neuron-CPD pairs; dashed line: chance-level performance). For each neuron with significant tonic spike-CPD coupling, a support-vector classifier was trained to select between 2 phase bins, one of which was centered around the preferred phase of tonic spike coupling, the other was centered 180˚ opposite. Decoding accuracy was cross-validated using 5 training-test splits. The distribution of decoding accuracy was significantly greater than chance for all timescales (Wilcoxon rank-sum test for each timescale, *p*-value denoted by asterisks, * for $p \leq 0.05$, ** for $p \leq 0.01$, *** for $p \leq 0.001$). **(B)** Same as (A), but for CPD phase decoding using bursting. For each neuron with significant burst-CPD coupling, a support-vector classifier was trained using bursting activity, and the phase bins were centered around the preferred phase of bursting.
(TIFF)

## Acknowledgments

We would like to thank M. Sotgia for lab management and support with animal handling and histology, Y. Bauer, G. Born, and A. H. Kotkat for their contributions to the data collections

and spike sorting, S. Schörnich for IT support, and B. Grothe for providing excellent research infrastructure.

## Author Contributions

**Conceptualization:** Davide Crombie, Christian Leibold, Laura Busse.

**Data curation:** Davide Crombie, Martin A. Spacek.

**Formal analysis:** Davide Crombie.

**Funding acquisition:** Davide Crombie, Christian Leibold, Laura Busse.

**Investigation:** Martin A. Spacek.

**Methodology:** Davide Crombie, Christian Leibold, Laura Busse.

**Project administration:** Christian Leibold, Laura Busse.

**Software:** Davide Crombie, Martin A. Spacek.

**Supervision:** Christian Leibold, Laura Busse.

**Visualization:** Davide Crombie.

**Writing – original draft:** Davide Crombie, Christian Leibold, Laura Busse.

**Writing – review & editing:** Davide Crombie, Christian Leibold, Laura Busse.

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
