## [Editor Report · Decision Letter 0]

21 Jul 2023

Dear Dr Busse, 

Thank you for submitting your manuscript entitled "Spiking activity in the visual thalamus is coupled to pupil dynamics across temporal scales" for consideration as a Research Article by PLOS Biology.

Your revised manuscript has now been evaluated by the PLOS Biology editorial staff as well as by the original academic editor, I am writing to let you know that we would like to send your submission back to the original reviewers.

However, before we can send your revised manuscript to reviewers, we need you to complete your submission by providing the metadata that is required for full assessment. To this end, please login to Editorial Manager where you will find the paper in the 'Submissions Needing Revisions' folder on your homepage. Please click 'Revise Submission' from the Action Links and complete all additional questions in the submission questionnaire.

Once your full submission is complete, your paper will undergo a series of checks in preparation for peer review. After your manuscript has passed the checks it will be sent out for review. To provide the metadata for your submission, please Login to Editorial Manager (https://www.editorialmanager.com/pbiology) within two working days, i.e. by Jul 25 2023 11:59PM.

Kind regards,

Luke

Lucas Smith, Ph.D.

Senior Editor

PLOS Biology

lsmith@plos.org

---

## [Decision Letter · Decision Letter 1]

28 Sep 2023

Dear Dr Busse,

Thank you for your patience while we considered your revised manuscript "Spiking activity in the visual thalamus is coupled to pupil dynamics across temporal scales" for publication as a Research Article at PLOS Biology. Your revised study has been evaluated by the PLOS Biology editors, the Academic Editor and by the original reviewer #3. Unfortunately, the original reviewers 1 and 2 were not available to assess this revision. In their place, we enlisted the help of a new reviewer, with relevant expertise, to assess the response to their comments (new reviewer #4). 

As you will see in their comments, the reviewers appreciate that the manuscript has been strengthened in the revision, and the find the study potentially interesting. However, they have raised a number of substantial new and lingering concerns, including with the data, statistics, and analyses, related to key conclusions in the study. We think that it would be essential to thoroughly address these concerns before we can consider your manuscript for publication at PLOS Biology. 

Given the interest in the topic, and after discussion with an Academic Editor with relevant expertise, we are willing to consider another revised version of the study that addresses the new round of reviews. Given the extent of revision needed, we cannot make a decision about publication until we have seen the revised manuscript and your response to the reviewers' comments. Your revised manuscript is likely to be sent for further evaluation by all or a subset of the reviewers.

**IMPORTANT - SUBMITTING YOUR REVISION**

*Re-submission Checklist*

*Published Peer Review*

*PLOS Data Policy*

*Blot and Gel Data Policy*

Sincerely,

Lucas

Lucas Smith, Ph.D.

Senior Editor

PLOS Biology

lsmith@plos.org

REVIEWS:

Reviewer #1: Did not re-review

Reviewer #2: Did not re-review

Reviewer #3: I would like to thank the authors for thoughtfully considering my questions and providing informative answers. Both the quality of the data analysis and the writing of the revised manuscript has been greatly improved. However, it still took a significant amount of efforts for me to read and digest the manuscript, due to the complexity of the data analyses, inadequate explanations of the methods, and typos in the text. Moreover, some of my previous questions were not adequately addressed or given clear explanations.

Major comments: 

1. I am convinced now that the multiple timescale analysis of pupil dynamics provides valuable new insights into the modulation of dLGN visual processing by behavioral state, and there is something interesting and robust in the coupling of the firing mode with the phase of pupil dynamics. However, I am still not sure about the reliability of the phase coupling for individual neurons or the decomposition of pupil dynamics by empirical mode decomposition (EMD). For example, if the segment data from each recording is separated into two halves, and the CPD and coupling analysis is applied to each half, shall we expect similar results regarding the timescale that gives the strongest coupling and the preferred coupling phase for burst vs. tonic firing at each timescale? Similarly, if the segment of data from the same unit is separated according to behavioral states, will similar coupling results be obtained across behavioral states? 

2. The authors now showed that the timescales to which units are most strongly coupled are diverse across mice and across sessions (Figure. S3C3 and S3C4). Although the diversity of dLGN neurons may be the reason underlying this result, another possibility for the result is a lack of reliability or specificity of the EMD analysis. 

3. In the answer to my original question 1, the authors refer to Figure R2. But Figure R2 does not look to be relevant. I suspect the authors actually referred to Figure R3. The pupil size gradient should have both negative and positive values. But it's unclear whether the normalized pupil gradient in Figure R3 was based on the absolute gradient, or whether 0 and 1 corresponded to the most negative and most positive values. Moreover, I asked about the relationship between the firing mode and the change of pupil sizes. Figure R3 showed the dLGN spike rate (spike count in 250 ms bin?) but not the firing mode. 

4. In Figure R7, no stats were performed to compare the distributions of the preferred phase under gray screen vs. sparse noise stimulation conditions. The preferred phase of the bursting firing under the gray screen condition actually looks to have a larger fraction concentrated at 0 deg phase, different from that under the sparse noise stimulation condition. 

Other minor comments: 

1. Line 74-75: 'However, we also noticed that similar ranges in pupil size fluctuations occurred within both active and quiescent behavioral states (Figure S1B3). ' Please provide a movie clip of the pupil recording and the trace of simultaneously recorded locomotion speed for the period when pupil reached the minimal area in the displayed range in the figure and when the animal was running. 

2. Possible typos: 

Line 161: Figure S1E was referred to in the text. But there is no Figure S1E. 

Line 168 -169: Figure S3C2 and S3C3 should be S3C3 and S3C4. 

Line 171: Figure S2C2 and S2C3 should be S3C3 and S3C4. 

New Reviewer #2 (who assessed the responses to Reviewers 1 and 2): The study investigates in the mouse the interplay between neural activity in the visual thalamus and pupil dynamic. The pupil controls retinal illumination in response to changing light levels but also reflects behavioral variables such as arousal and locomotor activity, which correlate with neuromodulatory processes critical for sensory processing. The paper argues that spontaneous visual thalamic activity shows multiscale temporal coupling to pupil dynamic, occurring preferentially at specific temporal phases across various frequencies. Tonic spikes and bursts show opposite phase preferences, with activity located preferentially at the peaks and troughs of the pupil the size dynamic (the effects are more pronounced for bursts than tonic spikes). This coupling and phase tuning is observed for diverse time scales in constant illumination, darkness, and upon visual stimulation. The temporal relationship is robust and not explained by behavioral variables such as locomotor activity and saccadic eye movements and is proposed as an organizing principle of visual thalamic activity.

Overall, this interesting, high-quality study adds to our understanding of how visual neurons respond during behavior. While the investigation is focused on ongoing activity, the paper does examine responses to naturalistic movies, exploring the interplay between ongoing and visually-evoked activity.

Although the analytical methods are involved, the paper is well written, and the results clearly presented. The authors have gone to great lengths to address the reviewers' concerns by adding new analyses and data and making comprehensive textual changes to the manuscript. The authors' response to the reviewers' is overall satisfying. However, issues of analysis and interpretation remain from the original submission and new issues were added in course of the new revision. 

Main comments:

1. The manuscript has repeated mentions of 'nested' temporal dynamics. However, readers may not be familiar with this concept (I am not). The term is used in reference to arousal in the Introduction (line 53) but is not defined. There is a sentence in results (lines 153-154) referring to nested modulations, but again, it is unclear what is meant. If the word is to be used, the authors should clearly explain it in the Introduction and explain what aspects of the data support the hypothesis (or provide evidence against it).

2. Multiple statements are also made about neurons being tuned to fewer or more time scales in different experimental conditions. For instance, Fig. 4 shows eye movements and quiescence don't influence coupling, but locomotion seem to have a significant effect. How should these differences be interpreted within the proposed framework of nested modulations? 

3. At a practical level, how sensitive is the analysis to the number of events detected? The bursts are very sparse, which could limit the potential for observing coupling at fast temporal scales. Are the differences in coupling and number of coupled frequencies reported for bursts and spikes (e.g., Fig. 1E-F) driven by the relative rarity of bursts? Would the analyses of the coupling of tonic spikes yield the same results if the number of spikes and bursts were matched? If the blue and red curves cannot be compared because of limitations in the data set, this should be stated explicitly.

4. Does the autocorrelation of spike/bursts contribute to the coupling across temporal scales? How are the results influenced by the neurons' spike-generation mechanism? Bursts and (tonic) spikes have distinct firing thresholds and, thus, by definition, occur at distinct time points. Does that contribute to the opposite temporal phase preferences? Bursts occur following periods of hyperpolarization, which must last 100 ms but can be of arbitrary long duration and thus intuitively could show correlation over a range of scales. While an analysis is provided for correlation across scales in the pupil size signal, I have not seen the same for the neural firing. This seems important for the interpretation of the biological interpretation but also to ensure that the results are not due to the spike/burst classification method.

5. The data in darkness is a great addition, but the infrared wavelength used for eye tracking in darkness is not specified. The mouse retina has low infrared sensitivity, but we have observed sensory responses in the brain in the 'dark-adapted' state at 700-750 nm. The methods should clearly state what measures were taken to ensure complete darkness. While the estimated firing rates as a function of pupil size in darkness (S1C1) resemble those in constant illumination (Fig. 1), it does not seem to be the case for Fano factors, which show distinct distributions. 

6. The data on movie responses are an interesting addition to the manuscript, but there are issues with the analysis and interpretation of the data. The example raster plot in Fig 5A1 clearly shows how the neuron's response to the visual stimulus toggles between states, showing distinct visual response patterns correlated with changes in pupil size (Fig. 5B1). The marginal plot on the right-hand side, which shows the sum of spike responses over the entire movie, ascribes the trial-to-trial variations in the total number of spikes entirely to noise. However, from the raster plot, the temporal responses to the visual stimulus seem reproducible within a consecutive block of trials. Some of the variations in firing rates over the stimulation epoch may, therefore, not reflect noise but changes in the neurons' visual receptive field (e.g., Aydin et al. 2018, Franke et al. 2022). Are these visual influences taken into consideration in the noise and SNR measurements? 

7. The state-dependence of visual receptive fields has implications for the subsequent decoding analyses. Since the pupil state changes the visual responses, how can we be sure that the decoding reflects the dynamic of ongoing activity? 

8. The paper has very few illustrations of raw data. 

a. Except for Fig. 5A, no burst spike raster plots are shown for the different visual illumination and stimulation conditions (bursts are barely visible in Fig. 5A, and tonic spikes are invisible in Fig. 2B). Firing rates in Figure 1C1 stem from a model fit of the data. While the fits may be perfect, the data is indirect, and fit quality is not reported. If the firing events in the neurons show a diversity of phase preferences, then neurons should fire bursts at distinct time points. 

b. The pupil size data in Fig. S1A is a great addition, but the measurements are normalized to the

---

## [Decision Letter · Decision Letter 2]

6 Feb 2024

Dear Dr Busse,

Thank you for your patience while we considered your revised manuscript "Spiking activity in the visual thalamus is coupled to pupil dynamics across temporal scales" for consideration as a Research Article at PLOS Biology. Your revised study has now been evaluated by the PLOS Biology editors, the Academic Editor and by the previous reviewers. The reviewer comments can be found at the end of this email. 

The reviewers have commented that the revision has addressed many of their previous comments - but they each detail a number of lingering concerns, noting that important parts of the response to reviewers have not been included in the revised manuscript, that additional clarifications are needed, and that the manuscript needs to be edited to make the study more broadly accessible and to remove jargon. While we think that the reviewer comments can largely be addressed without further experiments, we must emphasize that these last issues will need to be thoroughly resolved in order for us to consider the study further for publication.

In light of the reviews, we are willing to grant one last opportunity for you to thoroughly address the remaining points from the reviewers in a revision that we anticipate should not take you very long. We will then assess your revised manuscript and your response to the reviewers' comments with our Academic Editor and likely with the reviewers.

**IMPORTANT - SUBMITTING YOUR REVISION**

*Resubmission Checklist*

*Published Peer Review*

*PLOS Data Policy*

*Blot and Gel Data Policy*

Sincerely,

Lucas

Lucas Smith, Ph.D.

Senior Editor

PLOS Biology

lsmith@plos.org

REVIEWS:

Reviewer #1: I'd like to thank the authors for performing additional analysis to address my comments and concerns. I suggest to include Reviewer Figures R1 C-E and R4 in the manuscript as supplementary materials for readers to assess the data variability and conclusions. 

My other comments below mostly involve text changes. 

1. To motivate the multi-timescale arousal modulation of dLGN spiking activity, the authors highlighted that 'pupil size provided only limited information about firing rate: variance of firing rates within pupil size bins was high and in fact often larger than the mean (Figure 1C2; median Fano factor = 2.0' (Line: 86-87 ). However, the phase coupling strengths across a broad range of temporal scales were all notably small (with an R^2 of ~0.02 for burst firing and even smaller for tonic firing, within a theoretical range of R^2 from 0 to 1). Moreover, the variance of the phase coupling may also be large. If Fano factors were computed for bursts and tonic spikes at each phase, the resulting Fano factors would likely be substantial, just as what was shown in Figure 1C. It remains challenging to determine whether pupil size analysis already captured the majority of the modulation while multi-timescale analysis provided supplementary information, or vice versa. Therefore, it is essential not to diminish the significance of much simpler pupil size analysis (including analysis of both the absolute value of pupil sizes and pupil size gradients) in relation to dLGN spiking activity. In fact, the multi-scale pupil dynamic analysis may help to more closely capture the pupil size and pupil size gradient on both local and global temporal scales. 

2. Line 218-219: I am not convinced the conclusion that 'locomotion-dependent modulations appear unrelated to the phase coupling reported above' can be drawn based on the observation that 'significant coupling for tonic spikes was equally likely regardless of whether the component was correlated to locomotion or not', as for those that had correlated locomotion and CPD components, the locomotion can still be related to phase coupling.  

3. Line 210: Figure S1B4 does not exist. 

Reviewer #2: Most of my comments and concerns have been addressed satisfactorily. This is an extensive revision which has greatly improved the statistical analysis and demonstrate the robustness of the results. I have a few remaining minor concerns about the writing and interpretation of the data

General comments

1. In comment 1, Reviewer 1 asked about the reliability of the phase coupling measurement in individual neurons. In their response, the authors show convincingly that similar phase and frequency preferences are observed at the population distribution level across subsamples of the data but I could not find the data for the individual neurons' across subsamples. This is important because there is a repeated claim of diversity of coupling within the population, but because the results vary across sessions and mice this can only be shown by examining the coupling results for simultaneously recorded neurons from a same animal and experimental session across data subsamples. If I am not mistaken, no such data is presented.

2. In their response to my comments, the authors claim that the differential coupling of spikes and bursts cannot be explained by the neurons' intrinsic properties such as the spike generation mechanism. I don't quite follow the argumentation. However, as the new example cell in Figure 5 makes abundantly clear, bursts in the LGN strictly occur during periods of low level of single spike activity which, intuitively, could explain the oppositive phase preferences of bursts and single spikes. This hypothesis that could be addressed using computer simulations. If not, the authors should address the topic in Discussion and explain how this intuition is incorrect.

3. Some additional points or questions addressed in the response to reviewers, seem not to have been considered or integrated in the manuscript revision. I would consider integrating these data and analyses to the manuscript, at least in the Discussion or as supplementary material.

4. Despite comments from the reviewers, the Results and the figure captions still contain jargon (e.g. '(characteristic) time scales', 'characteristic frequency', 'phase synchronization', 'phase-coupling', 'phase-coupling framework'). These terms are introduced without explanation or in way that is not adapted to the broad readership of the journal. The authors should screen for such words, add definitions in place (not buried in the Methods) or replace them with concrete explanations or simpler terms.

5. Along the same lines, the wording of the findings and conclusions in the abstract and in the manuscript are phrased in a very technical language that is inaccessible to 99% of biologists. For a journal like PLOS Biology, the findings should be conveyed in more simple language (e.g. "single spikes and bursts of action potentials in the visual thalamus preferentially occur at distinct, approximately opposite phases of the pupil behavior indicating …") to be accessible to biologists from a different field (e.g. molecular or developmental biology).

6. With the revisions, the Results has become somewhat bloated with methodological sentences. I recommend moving these sentences to Methods to improve readability.

7. It is good practice to include in Discussion a paragraph on limitations of the study. This seems to be missing. 

Details:

L64 First sentence does not seem to encompass the overall goal of the study.

L67 It would make sense to mention that pupil size variations also change retinal illumination, which can also affect the dynamics of visual responses.

L93 The firing events in 2 seem to occur in absence of pupil fluctuations. Is this an example of a non-pupil driving firing event or are their fluctuations that are too small to be resolved?

L94 It would make sense to point out how these different types of firing events could reflect different types of modulations.

Figure 1: panel D1 is difficult to parse. The repeated gray lines that overlap with the colored lines do not really help to understand the decomposition. I understand that the authors would like to highlight the correlation but what is salient are the deviations. I would rather plot gray and colored lines separately showing how colored lines represent a decomposition of the gray line by putting them next to each other.

Figure 1: D2 caption uses both the terms "characteristic frequency" and "characteristic timescale". Only the latter is defined in Methods. Pick one and use throughout the manuscript. Furthermore, these terms are not introduced in Results. Either include in Results or discard.

L97: "beyond the size" what the authors means is "absolute, instantaneous size".

L104: missing reference for "empirical mode decomposition".

L105: The text "capturing its underlying dynamics" does not seem to be adequately supported. The motivation for choosing this specific decomposition is not explained. No reasoning explaining how this decomposition is more appropriate than others is presented (e.g. Fourier decomposition, wavelet transform, etc). The time sequences in Figure 1D1 seem to suggest otherwise as to the eye only one of the components seems strongly correlated with pupil size. Furthermore, the absolute measure of explained variance and absolute correlations are not provided. 

Figure 1D3 the caption is not clear. What are the gray dots? A box plot cannot show 'a distribution'.

Line 108-110: Figure 1D3 is cited to support the claim that components were found/extracted but no data on effect size and significance levels is shown. Some of the datapoints lie on the abscissa. Presumably these data points are not significant.

Line 155-156: The statistical testing for this paragraph is not clear. Which threshold was used to test the significance of coupling modulations at multiple time scales. Were the results corrected for multiple comparisons? Perhaps the answer lies in the following sentences but the statistical tests and sample sizes supporting these statements are not clear.

L168-171: Speculation. Does not belong to Results. The source of the variability across mice or recording sessions is not directly addressed. 

L305: Variability

---

## [Editor Report · Decision Letter 3]

5 Apr 2024

Dear Laura,

Thank you for the submission of your revised Research Article "Spiking activity in the visual thalamus is coupled to pupil dynamics across temporal scales" for publication in PLOS Biology, and thank you for addressing the last reviewer and editorial requests. On behalf of my colleagues and the Academic Editor, Jonathan Demb, I am pleased to say that we can in principle accept your manuscript for publication, provided you address any remaining formatting and reporting issues. These will be detailed in an email you should receive within 2-3 business days from our colleagues in the journal operations team; no action is required from you until then. Please note that we will not be able to formally accept your manuscript and schedule it for publication until you have completed any requested changes.

**IMPORTANT: Please note - I have updated your data availability statement to reference the dataset that you provided me: https://gin.g-node.org/laura_busse/pupil_timescales_dLGN. Please do double check that this all looks correct. Also, as mentioned over email, please generate a DOI for this dataset, and add it to the data availability statement in our online system, as this will be needed for publication. 

PRESS

Sincerely, 

Luke

Lucas Smith, Ph.D.

Senior Editor

PLOS Biology

lsmith@plos.org